# Electrochemically induced hyperfluorescence based on the formation of charge-transfer excimers

Chang-Ki Moon[1,2], Yuka Yasuda[3], Yu Kusakabe[3], Anna Popczyk[1], Shohei Fukushima[3], Julian F. Butscher[1], Nachiket Pathak[1], Joel Schlecht[4], Kuraudo Ishihara[3], Oliver Dumele [4], Hironori Kaji [3] ✉ & Malte C. Gather [1,2] ✉

Used extensively in sensing applications, the application of solution-state electrochemiluminescent devices (ECLDs) in lighting and displays has been constrained by their low luminance and short operational lifetime. Here, we introduce ECLDs based on electrochemically induced hyperfluorescence (ECiHF), and demonstrate their use in a calligraphic display. We use the double-decker arrangement assumed by the electron donor and acceptor segments of the molecule TpAT-tFFO to realize thermally activated delayed fluorescence from an electrogenerated charge-transfer excimer state. ECLDs based on this strategy achieve improved efficiency, a luminance of >6200 cd m$^{-2}$ and their operational lifetime is more than 10-fold longer than all previous ECLDs with meaningful efficiency or brightness. Using spectro-electrochemical analysis, we identify energy level alignment between excimer and emitter as a crucial factor for efficient ECiHF. Our findings highlight the potential of ECiHF for improving ECLDs and pave the way to commercial applications of this form of fluid light.

By combining electrochemistry with light emission, electro-chemiluminescence (ECL) has become a versatile tool in biomedical research, particularly for sensitive detection of biomolecules in immunoassays and diagnostics[1–3], food safety analysis[4], and environmental surveillance[5]. Unlike other types of electrically generated light emission that rely on charge carrier transport and recombination in solid state, such as organic light-emitting diodes (OLEDs)[6] and light-emitting electrochemical cells[7,8], ECL generally occurs by ionic reactions in liquid or gel state, thus providing great flexibility in device form factor and greatly simplifying fabrication[9–12]. To extend the ECL technology to future applications in lighting and display applications, and hence to further benefit from the cost-effectiveness and scalability of this fluid light, recent research has explored novel luminophores, electrode configurations, and operating mechanisms to enhance the brightness, efficiency, and operational stability of electro-chemiluminescent devices (ECLDs)[13–18]. Achieving intense ECL generally necessitates production of abundant radical ions and their subsequent rapid radiative recombination. However, the accumulation of radical ions can trigger side reactions that deteriorate the luminophores in the device[19], limiting the maximum brightness of ECLDs that rely on this annihilation process to less than 700 cd m$^{-2}$ and restricting their operational times to a few minutes at most.

In OLEDs, luminophores supporting thermally activated delayed fluorescence (TADF) have gained great popularity as they enhance the fraction of emissive singlet excitons to ~100% by rapidly depopulating non-emissive triplet states via reverse intersystem crossing (RISC)[20,21]. In ECLDs, TADF emitters have also been demonstrated to achieve up to a 4-fold improvement in ECL efficiency[22–24]; however, so far with

[1]Humboldt Centre for Nano- and Biophotonics, Institute for Light and Matter, Department of Chemistry and Biochemistry, University of Cologne, Greinstr. 4-6, 50939 Köln, Germany. [2]Organic Semiconductor Centre, School of Physics and Astronomy, University of St Andrews, North Haugh, St Andrews KY16 9SS, United Kingdom. [3]Institute for Chemical Research, Kyoto University, Gokasho, Uji, Kyoto 611-0011, Japan. [4]Institute of Organic Chemistry, Department of Chemistry and Biochemistry, University of Cologne, Greinstr. 4-6, 50939 Köln, Germany. ✉e-mail: kaji@scl.kyoto-u.ac.jp; malte.gather@uni-koeln.de

limited benefit to the maximum achievable luminance and operational stability. Using TADF emitters in ECLDs also presents additional challenges, particularly related to solvent selection, as positive solvatochromism leads to significant spectral shifts and broadening of the emission in high-polarity solvents[25,26]. Hyperfluorescence (HF), which combines a conventional fluorescent emitter with a triplet sensitizer[27], offers a promising alternative that enables high luminance and efficiency without introducing substantial spectral shifts. In our previous exciplex-based ECLDs[28,29], the exciplex state served as a triplet sensitizer[30,31]; however, the RISC process on the exciplex was not sufficiently fast to support efficient HF.

In this paper, we realize electrochemically induced hyperfluorescence (ECiHF) through the electrogeneration of charge-transfer (CT) excimers−rather than exciplexes−, thus achieving rapid RISC and subsequent energy transfer to a fluorescent emitter. Our study investigates two TADF molecules, the recently reported TpAT-tFFO[32] and TpATtBu-tFFO, a previously unreported analogue; both feature face-to-face alignment of the donor and acceptor components, thus exhibiting excellent TADF characteristics. Using concentration-dependent photoluminescence (PL) spectroscopy in solution, we observe a transition from monomer emission to excimer emission while preserving clear hallmarks of TADF as the concentration of TpAT-tFFO increases. Adding a rubrene dye to these high-concentration TpAT-tFFO solutions then leads to the appearance of HF and enables ECiHF in AC-driven ECLDs, yielding simultaneously high luminance, efficiency, and operational stability. To showcase the benefits of this enhanced light emission, we demonstrate an ECL display featuring a calligraphic electrode design. Finally, to gain insights into molecular design rules for future high-performance ECLDs, we compare the excimers of TpAT-tFFO and TpATtBu-tFFO. The latter has a higher energy and shows lower performance in ECLDs with a stronger dependence on operating voltage and frequency. Absorption spectroelectrochemistry confirms excimer formation as the primary operating mechanism in TpAT-tFFO devices, and reveals a mixed operating mechanism for TpATtBu-tFFO devices, in which excimer formation and ionic annihilation compete.

## Results

### Electrochemically induced hyperfluorescence mechanism

A schematic illustration of the ECiHF mechanism is given in Fig. 1a. By alternately between positive and negative voltages at the electrodes, oxidized and reduced forms of the TADF molecules accumulate near each electrode surface. These redox products recombine to electrostatically bound CT excimers, with the highest occupied molecular orbital (HOMO) and lowest unoccupied molecular orbital (LUMO) localized on donor and acceptor segments of the TADF molecules, respectively. This spatial separation results in minimal orbital overlap and thus enables efficient intermolecular CT transitions. The CT excimers undergo intersystem crossing (ISC) and RISC cycles, and transfer singlet energy to neighboring fluorescent emitters via FRET, thereby achieving an exciton utilization exceeding the 25% spin-statistical limit.

Figure 1b shows three-dimensional representations of the molecular structures of TpAT-tFFO and TpATtBu-tFFO. The triptycene linkers in these compounds enable a face-to-face alignment of the donor (dimethyl dihydrouridine) and acceptor groups (either diphenyl triazine or di-tert-butyl triazine), with the HOMO and LUMO localizing on the donor and acceptor groups, respectively. The double-decker structures are further expected to facilitate intermolecular π-π stacking at high concentrations. Consequently, these molecules are expected to allow intramolecular[32] as well as intermolecular CT transitions.

The concentration-dependent PL spectra of TpAT-tFFO solutions in a 2:1 by volume mixture of toluene and acetonitrile showed a shift in emission from blue fluorescence (482 nm) at <0.3 mM to green fluorescence (530 nm) at >30 mM (Fig. 1c and d). The gradual red-shift indicates the transition from monomer to excimer emission, with the saturation in spectral shift at 30 mM indicating that pure excimer state is reached at this concentration. According to our previous TD-DFT calculations for TpAT-tFFO[32], the energy level of the singlet CT state increases as the donor-acceptor separation increases. A reduction in energy by 0.23 eV corresponds to a decrease in separation from 4.72 Å (intramolecular CT state) to 4.00 Å (intermolecular CT state), driven by strong intermolecular electrostatic binding. Our simulation model, using molecular dynamics simulations and time-dependent density functional theory calculations, also yielded cation-anion aggregated pairs of TpAT-tFFO with lower singlet-state energies in the intermolecular CT transition than in the intramolecular CT transition in a toluene:acetonitrile mixture (see Supplementary Table 1 and Supplementary Fig. 1).

Figure 1e shows the absorption (spectrally resolved molar extinction coefficient) and emission spectra of TpAT-tFFO at concentrations of 0.1 mM and 10 mM, measured in a toluene:acetonitrile mixture and in pure toluene, respectively. The red-shift in emission spectrum is significant in toluene:acetonitrile but the shift is only 7 nm in pure toluene, indicating that the aggregation is facilitated by adding acetonitrile, a high polarity solvent. The aggregation in the ground state is negligible, given that the absorption spectrum of TpAT-tFFO remained nearly unchanged regardless of mixing solvent or increasing concentration.

Transient PL measurements indicate TADF is present across all concentrations tested here (Fig. 1f). Furthermore, variable-temperature PL shows the increased ISC and RISC rates at higher temperature for both monomer and excimer emissions (see Supplementary Fig. 2). The ratio of the photoluminescence quantum yield (PLQY) of the delayed and prompt emission ($\Phi_d/\Phi_p$) was higher for the excimer emission at high concentrations than for the monomer emission, while the ISC-RISC processes showed comparable rates (see Fig. 1g and Supplementary Table 2). Adding 10 mM 2,8-di-tert-butyl-5,11-bis(4-tert-butylphenyl)−6,12-diphenyltetracene (TBRb) to an 80 mM excimer solution of TpAT-tFFO yielded pure HF, with prompt and delayed fluorescence lifetimes of 25 ns and 172 ns, respectively (Fig. 1h) and no sign of residual excimer emission in this mixture (see Supplementary Fig. 3). The HF is facilitated by the very fast RISC on the excimer (rate constant of $1.2 \times 10^7 \, s^{-1}$) and also by rapid FRET.

### Device performance

Adding a supporting electrolyte to the hyperfluorescent solution, we fabricated two types of ECLDs in parallel-electrode (PE) configuration (Fig. 2a). The first, a glass-glass device, uses two ITO-coated glass substrates, separated by a 30-micrometer gap filled with the solution and had four square-shaped emissive surfaces measuring 4 mm² each (see Methods for detailed fabrication process). The intensity of light emitted through both substrates was equal due to the optically symmetric structure of the device; luminance values quoted in the following refer to the light emission to one side only. The second configuration, a glass-mirror device, includes a silver mirror with a passivation coating on the outer side of one of the glass substrates to reflect the emission back into and through the device and thus achieve unidirectional and therefore brighter emission through one of the substrates.

Figure 2b shows the ECL spectra of these two devices; both spectra peak at 570 nm, clearly indicating that light emission is exclusively from TBRb. By contrast, devices without TBRb showed direct ECL from the electrogenerated CT excimer; see Supplementary Fig. 4. The glass-mirror device has increased self-absorption by TBRb during the light recycling due to a partial overlap between the ECL spectrum and the absorption spectrum of TBRb. As a result, the emission intensity of the device at wavelengths shorter than the peak wavelength is slightly reduced, leading to a narrower emission spectrum. Figure 2c shows the root-mean-square (rms) current density-

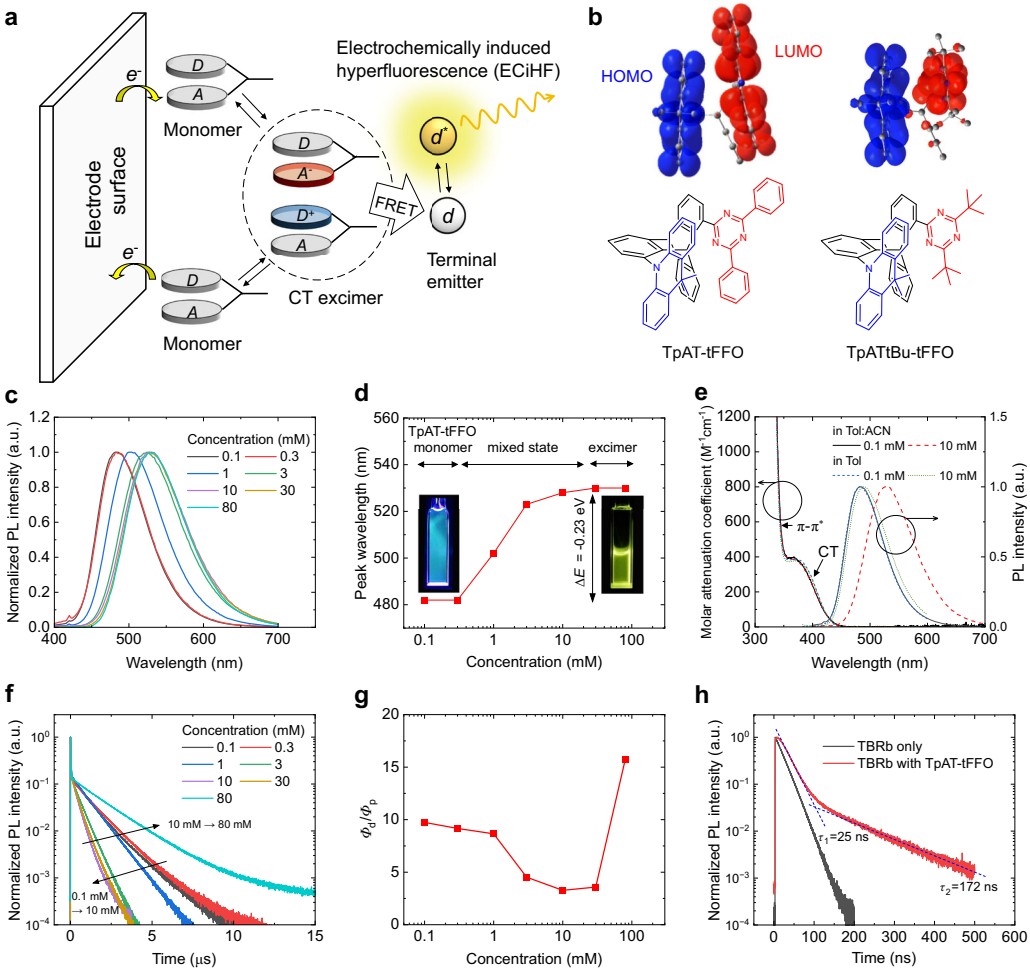

**Fig. 1 | Electrochemically induced hyperfluorescence (ECiHF). a** Schematic of ECiHF under AC operation with formation of CT excimers and energy transfer to the terminal emitter. **b** Molecular structures of TpAT-tFFO and TpATtBu-tFFO in three-dimensional configurations, with the donor and acceptor groups represented as blue and red segments, respectively, in the line drawings. For both molecules, the HOMO (blue surface) and LUMO (red surface) states are separated on the donor and acceptor groups, respectively. **c** Photoluminescence (PL) spectra of TpAT-tFFO in a 2:1 by volume mixture of toluene and acetonitrile. **d** Peak wavelengths of PL spectra for TpAT-tFFO at concentrations ranging from 0.1 mM to 80 mM. **e** Molar attenuation of TpAT-tFFO at concentrations of 0.1 mM and 10 mM, measured in a toluene:acetonitrile mixture and pure toluene. **f** Transient PL of TpAT-tFFO at concentrations ranging from 0.1 mM to 80 mM. **g** Ratio of the quantum yield of delayed to prompt fluorescence ($\Phi_d/\Phi_p$) against TpAT-tFFO concentration. **h** Transient PL of solution containing 10 mM of TBRb and 80 mM of TpAT-tFFO, compared to solution containing 10 mM of TBRb only.

voltage-luminance (*JVL*) characteristics of both devices under AC operation at a frequency of 190 Hz. Both devices showed very similar current-voltage characteristics due to their identical electrochemical nature. The glass-glass device achieved a luminance of 3650 cd m$^{-2}$ at $V_{rms}$ = 4.0 V, while the glass-mirror device showed a 1.70-fold enhancement, reaching a maximum luminance of 6220 cd m$^{-2}$. The absolute ECL quantum efficiency ($\Phi_{ECL}$) and luminous efficacy (LE) shown reached maximum values of 0.56% and 1.66 lm W$^{-1}$, respectively, for the glass-glass device (only considering the emission in one direction), and 0.93% and 2.66 lm W$^{-1}$ for the glass-mirror device (Fig. 2d). The glass-mirror device thus showed an improvement in luminance, LE, and $\Phi_{ECL}$, by 1.60 to 1.70-fold. This enhancement, while significant, was less than the theoretical improvement of 1.97 expected when considering the reflectivity of an ideal silver surface. The loss is likely due to differences in the ITO surface condition between the two devices as the UV-ozone treatment of ITO had to be reduced from the optimum duration of 15 min to 3 min post-coating of the silver mirror, which in turn will affect the faradaic process during device operation. Self-absorption by TBRb further contributes to the lower-than-expected brightness in the glass-mirror device.

Figure 2e shows estimates of LT$_{50}$ of our glass-glass ECLDs, obtained by measuring the time until the ECL intensity decreases to 50% of the maximum value when continuously operated under AC driving (see luminance vs time curves in Supplementary Fig. 5). Under potentiostatic (voltage-controlled) operation, the LT$_{50}$ values were 369 and 60 seconds at initial luminance levels of 100 cd m$^{-2}$ and 1000 cd m$^{-2}$, respectively. Under galvanostatic (current-controlled) operation, the LT$_{50}$ values for these initial luminance levels improved to 1252 and 96 seconds, respectively.

Table 1 summarizes the performance of the ECLDs in this study and compares them to prior works using the same PE configuration with ITO electrodes but different operating mechanisms, i.e. exciplex formation and ionic annihilation. Devices based on excimer formation outperformed annihilation-based devices in luminance as they use a more efficient bimolecular recombination process. The excimer-based device achieved a major further improvement in luminance, $\Phi_{ECL}$, and operational lifetime over the exciplex-based device. By further using reflection of back-emission, the glass-mirror device achieved a further increase in luminance, reaching an unprecedented ECL luminance of 6220 cd m$^{-2}$. Galvanostatic operation enhanced the LT$_{50}$ lifetime relative to potentiostatic operation, due to the better charge carrier

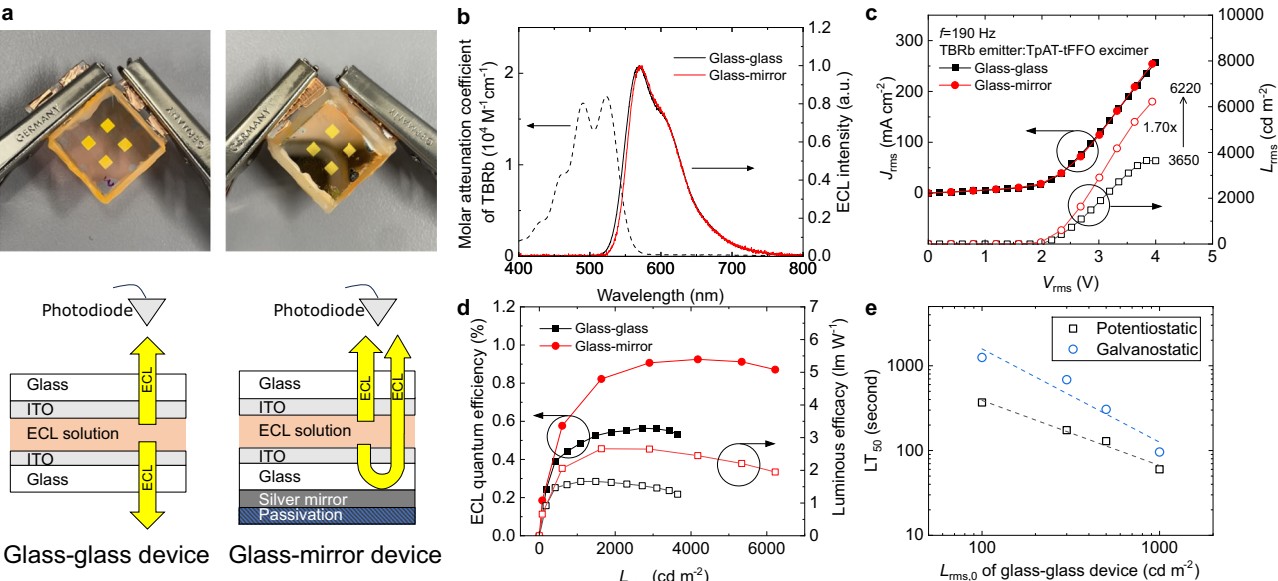

**Fig. 2 | ECLDs operating based on the ECiHF mechanism using TpAT-tFFO as excimer-forming material and TBRb as terminal emitter. a** Photographs and structures of glass-glass and glass-mirror parallel electrode (PE) device configurations with 4 pixels, each measuring 2 mm by 2 mm, operated simultaneously. **b** ECL spectrum of each device partially overlapping with molecular attenuation coefficient of TBRb. **c** Root-mean-square (rms) current-voltage-luminescence (JVL) characteristics. **d** Quantum efficiency and luminous efficacy. Data obtained under AC operation at a frequency of 190 Hz. **e** Operational lifetime LT$_{50}$ of the glass-glass device under potentiostatic and galvanostatic operation, respectively. All data is based on emission to one side of the substrate only.

**Table 1 | Summary of characteristics of ECLDs with PE configuration using ITO electrodes, AC driving, and different operating mechanisms**

| Operating mechanism | Material | Medium | L$_{max}$ (cd m$^{-2}$) | LE$_{max}$ (lm W$^{-1}$) | $\Phi_{ECL}$ (%) | LT$_{50}$ (sec), L$_0$ ~ 100 cd m$^{-2}$ |
|---|---|---|---|---|---|---|
| CT excimer (this work) | TBRb & TpAT-tFFO | toluene-acetonitrile | 3650, 6220 (M) | 1.66, 2.66 (M) | 0.56, 0.93 (M) | 1252 |
| Exciplex[28,29] | TBRb, TAPC & TPBi | toluene-acetonitrile | 1250, 2260 (T) | 1.17, 2.06 (T) | 0.38 | 75 |
| Annihilation[28] | TBRb | toluene-acetonitrile | 390 | 0.74 | – | 26 |
| Annihilation[18,35,39] | Ru(bpy)$_3^{2+}$ | ionic liquid | 700 | 0.20 | 0.08 | 300 |
| Annihilation[34] | Ru(bpy)$_3^{2+}$ & TiO$_2$ nanoparticles | propylene carbonate | 165 | – | – | 1000 |

*M* indicates a device with a mirror coating on one substrate side. *T* indicates a device with a mesoporous TiO$_2$ electrode. Reference [29] reports $\Phi_{ECL}$ values of 0.35% and 0.52% for operation at 300 Hz and 100 Hz, respectively; this table shows the $\Phi_{ECL}$ value at 300 Hz, at which the excimer devices showed higher luminance. Reference [18] measured the lifetimes at an initial luminance of approximately 115 cd m$^{-2}$. Reference [34] measured the lifetime at an initial luminance of 165 cd m$^{-2}$.

regulation which maintains ionic balance[18], and reduces local thermal stress due to local concentrations in charge flux at protruded regions of the slightly rough ITO surface[33]. As a result, the CT excimer-based device achieved an LT$_{50}$ value exceeding 1200 seconds at an initial luminance of 100 cd m$^{-2}$, to our best knowledge, surpassing the current record of 1000 seconds for a much dimmer ECLD using a Ru(bpy)$_3^{2+}$ luminophore with TiO$_2$ nanoparticles (where peak luminance was only 165 cd m$^{-2}$)[34]. We attribute these improvements mainly to two factors: (1) the presence of ECiHF with rapid ISC/RISC cycles on the CT excimer, and (2) the greater robustness of the excimer against chemical degradation at high-voltage and during prolonged operation.

Innovations in electrode structures have the potential to increase the operational lifetime of ECLDs operating through ECiHF even further. For example, Kang et al. utilized gold electrodes for ECLDs and employed Ru(bpy)$_3^{2+}$ luminophores for annihilation. These strategies effectively extended the LT$_{50}$ lifetime values to 81 min and 600 min in floating bipolar[18] and coplanar electrode arrangements[35], respectively.

### Calligraphic display

Next, we produced a simple ECLD display that leverages the enhanced brightness provided by ECiHF as well as the design freedom offered by ECL. Figure 3a shows the floating bipolar electrode (FBE)[18] configuration used for this display, with two parallel stripes of ITO on the top glass separated from 70-nm-thick gold electrodes on the bottom glass by a 30 μm tall gap that was filled with ECL solution. This architecture facilitates the display of calligraphy by fine-pattering of the gold; in the present example the gold electrodes were patterned with a resolution of 10 μm forming the words "Köln" (Cologne) and "京都" (Kyoto), the logos of the institutes involved in this study, and a tree branch motif. Applying AC voltage to the electrodes located along the edges of the substrate generated electrical double layers near the surfaces of the opposing electrodes, thereby triggering the ECL reaction. Figure 3b shows a photograph of the operating device, clearly displaying the calligraphy. Microscope images confirm that the light intensity profile closely follows the fine electrode pattern (Supplementary Fig. 6). By assigning a dedicated contact pad to each letter, individual letters can be selectively addressed and illuminated (Supplementary Movie 1).

### Excimer material with increased band gap

We also studied ECiHF operation using the second excimer-forming material, TpATtBu-tFFO, which has a higher band gap (3.25 eV) than TpAT-tFFO (2.86 eV) and also features TADF in thin films (see

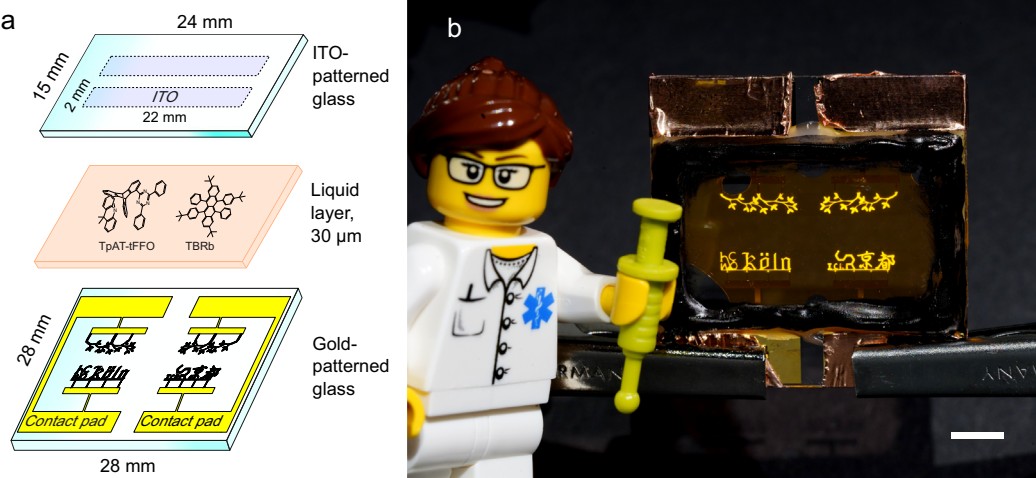

**Fig. 3 | Calligraphic ECLD display. a** Floating bipolar electrode configuration featuring calligraphic gold electrodes with a minimum feature size of 10 μm and non-patterned rectangular ITO electrodes that are at a floating potential. The overlap of the gold and ITO electrodes defines a total emissive area of 11.4 mm². **b** Photograph of device in operation with LEGO™ figure for size reference. Alligator clips supply AC voltage via contact pads on the left and right edges. The orange region in the center of the device is the area filled with the ECL solution. The scale bar at the bottom right represents a length of 5 mm.

Supplementary Figs. 7 and 8). Like TpAT-tFFO, TpATtBu-tFFO exhibited a concentration-dependent transition from monomer to excimer emission, shifting from deep-blue monomer emission at 442 nm to greenish-blue excimer emission at 484 nm (Fig. 4a). The presence of TADF was confirmed for both monomer and excimer emission using variable-temperature time-resolved PL measurements (see Supplementary Fig. 9). The TpATtBu-tFFO excimer emission showed a stronger overlap with TBRb absorption than the TpAT-tFFO excimer (see Supplementary Fig. 10), but a slower RISC rate of $7.2 \times 10^5\,\text{s}^{-1}$ (see Supplementary Table 3 and Supplementary Fig. 11).

Figure 4b compares the frequency-dependent ECL intensity of TpAT-tFFO and TpATtBu-tFFO based devices in a glass-glass PE configuration at different applied voltages. While the TpAT-tFFO device showed no shift in optimal frequency ($f_{\text{opt}} = 190\,\text{Hz}$), the TpATtBu-tFFO device exhibited a gradual upward shift. Additionally, the TpATtBu-tFFO device displayed S-shaped *JVL* characteristics, i.e. a simultaneous decrease in current density and luminance at $V_{\text{rms}} > 3.0\,\text{V}$, followed by an increase in both at even higher voltages (Fig. 4c). A similar S-shaped *JV* curve was previously observed in an ECLD based on annihilation between TBRb ions[28] when rapid degradation of TBRb molecules occurred. However, in contrast to the TBRb devices, our TpATtBu-tFFO device showed a subsequent increase in luminance at higher voltage. These observations suggest that the TpATtBu-tFFO device operates via a mixed mechanism involving both ionic annihilation and excimer-formation. Below $V_{\text{rms}} = 3.0\,\text{V}$, annihilation dominates and causes rapid degradation of TBRb molecules. At higher voltage, the excimer-formation process becomes effective, which increases the luminance again as this process does not involve TBRb ions. Nevertheless, the two processes remain in competition, with their relative contributions depending on the operation frequency. Higher frequencies led to increased luminance, with a maximum of 1640 cd m⁻² achieved at $f = 1\,\text{kHz}$.

**Absorption spectroelectrochemistry**

To gain deeper insights into the differences between TpAT-tFFO and TpATtBu-tFFO devices, we performed spectroelectrochemistry analysis[36] (see the setup in the Methods section and Supplementary Fig. 12). This technique tracks changes in optical density (ΔOD) as light passes through the active area of the ECLD with and without voltage application. When applying DC voltage to the TpAT-tFFO device without the terminal emitter TBRb (Fig. 5a), an absorption peak for

TpAT-tFFO cations was observed at 471 nm, along with a broad absorption band for anions ranging from 420 to 620 nm. The TpATtBu-tFFO device exhibited similar absorption bands, as shown in Fig. 5b, attributed to its structural similarity. For a device using solely TBRb (Fig. 5c), absorption peaks were detected at 424 nm for cations and 438 nm for anions, respectively. The bleaching of TBRb absorption at 490 nm and 525 nm (ΔOD < 0) is attributed to the reduced number of neutral TBRb molecules due to ionization.

Figure 5d shows the ΔOD curves of the TpAT-tFFO device under AC operation at $V_{\text{rms}} = 3.5\,\text{V}$ and frequencies of 190 Hz and 1 kHz (green and yellow lines), respectively. We simulated the ΔOD curves (blue and brown broken lines) using the cationic and anionic absorption spectra of TpAT-tFFO and TBRb, as well as the bleaching spectrum of neutral TBRb. At both frequencies, the ΔOD simulations include only the combined absorption from TBRb(+) and TBRb(−) along with TBRb bleaching, while no absorption from TpAT-tFFO(+) or TpAT-tFFO(−) was observed. This indicates that, as shown schematically in Fig. 5e, TpAT-tFFO ions swiftly return to their neutral state via rapid excimer formation followed by a swift FRET process, thus bypassing unnecessary electron transfer processes. This operation produces TBRb(+) and TBRb(−) by redox processes at the electrode surfaces. These ions tend to accumulate during device operation as the annihilation process between them is slower than the excimer formation. The formation of these long-lived TBRb excitons leads to bleaching of the TBRb absorption bands. The larger degree of bleaching observed at 190 Hz, compared to 1 kHz, is attributed to the increased formation of TBRb excitons, consistent with the frequency-luminance curve in Fig. 4b.

The simulation results for ΔOD of the TpATtBu-tFFO device (Fig. 5f), operating at $V_{\text{rms}} = 3.5\,\text{V}$, show an accumulation of TBRb(+) and TpATtBu-tFFO(−) ions along with TBRb bleaching at both tested frequencies. This indicates TBRb(−) and TpATtBu-tFFO(+) do not accumulate during device operation. As illustrated in the schematic in Fig. 5g, the different shape of the ΔOD curve is likely because of a HOMO-to-HOMO electron transfer from neutral TBRb to TpATtBu-tFFO(+) that proceeds at a rate comparable to or faster than the excimer formation. This leads to an overproduction of TBRb(+) and insufficient generation of TpATtBu-tFFO(+), which in turn results in an imbalance in ion concentration; TBRb(−) and TpATtBu-tFFO(+) are effectively depleted through annihilation and excimer formation, while their corresponding counter ions accumulate during device operation. Our spectroelectrochemistry data therefore confirms the mixed operation

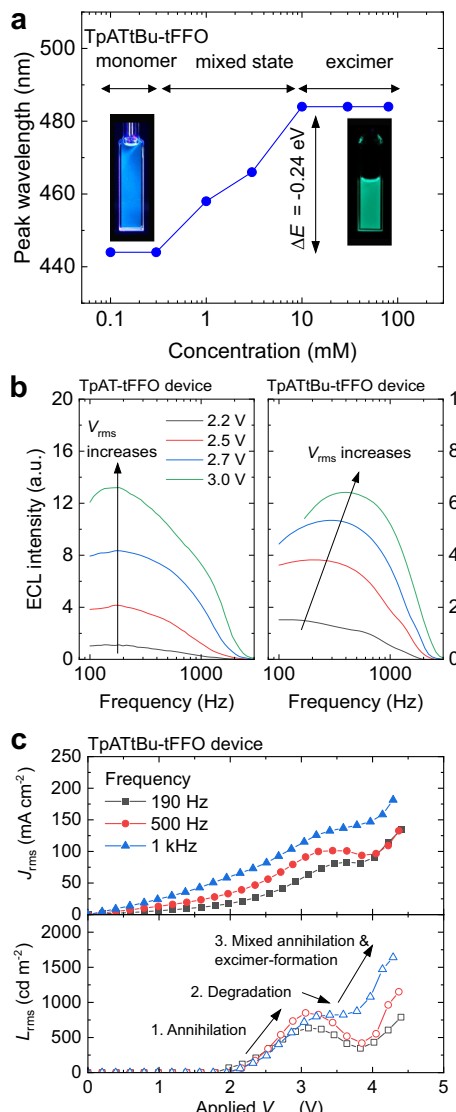

**Fig. 4 | ECLDs based on excimer formation in TpATtBu-tFFO. a** Shift in PL emission peak with concentration of TpATtBu-tFFO, demonstrating the transition from monomer to excimer emission. **b** Frequency-dependent ECL intensity at different applied voltages, comparing TpAT-tFFO and TpATtBu-tFFO devices. **c** Rms *JVL* characteristics of the TpATtBu-tFFO device operating at 190 Hz, 500 Hz, and 1 kHz.

mechanism of the TpATtBu-tFFO device. It can further be seen that the enhanced excimer formation process at higher frequencies, shown in the current-voltage-luminance curves in Fig. 4c, correlates with a decreased ΔOD from TpATtBu-tFFO(−) at a higher frequency.

The electron transfer process is facilitated by a significant energy level difference $\Delta E_{HH}$ between the HOMO levels of TpATtBu-tFFO and TBRb, namely $\Delta E_{HH} = 0.55$ eV. By contrast, electron transfer between TpAT-tFFO and TBRb was not significant, likely due to the smaller $\Delta E_{HH}$ of 0.39 eV. Similarly, the LUMO-to-LUMO electron transfer is not likely to be significant between TpAT-tFFO and TBRb as the energy level difference $\Delta E_{LL}$ is only 0.29 eV. Although $\Delta E_{LL}$ between TpATtBu-tFFO and TBRb is also large at 0.52 eV, we did not observe any evidence of LUMO-to-LUMO electron transfer in our experiments.

## Discussion

In summary, we explored the mechanism of ECiHF and demonstrated significant improvements in luminance, efficiency, and operational lifetime of ECLDs, enhancing their potential for lighting and display

applications. The TADF materials TpAT-tFFO and TpATtBu-tFFO formed CT excimers exhibiting strong delayed fluorescence at concentrations above 30 mM. Rapid RISC on the TpAT-tFFO excimer, followed by FRET to fluorescent TBRb, resulted in efficient hyper-fluorescence. An ECLD fabricated with two ITO-coated glass substrates achieved a maximum luminance of 3650 cd m⁻² in each direction, which increased to 6220 cd m⁻² upon adding a silver mirror to retro-reflect emission from one side of the device. Additionally, for an initial luminance of 100 cd m⁻², the $LT_{50}$ values exceeded 6 min under potentiostatic operation and 20 min under galvanostatic operation—representing the most stable ECLD performance reported to date. Combined with an FBE device structure, the enhanced brightness enabled by ECiHF, facilitated the realization of a calligraphic display with 10 μm resolution. These advances in performance are poised to open up opportunities for new device architectures and, in turn, new applications of this unique class of fluid-state light-emitting devices.

To shed light on the physics of the ECiHF process and to establish guidelines for material selection and performance optimization, we investigated the impact of energy level alignment between the excimer-forming materials and terminal emitter. We found that $\Delta E_{HH}$ and $\Delta E_{LL}$ below 0.4 eV is beneficial for efficient ECiHF. The device using TpAT-tFFO, where both gaps are below this threshold, exhibited a monotonic increase in *JVL*, an optimal operating frequency (190 Hz) independent of applied voltage, and a robust excimer-formation process unaffected by operating frequency. In contrast, the device using TpATtBu-tFFO, where $\Delta E_{HH}$ exceeds 0.5 eV, showed S-shaped and frequency-dependent *JVL* curves and an upward shift in the optimal frequency with increasing voltage. Both likely arise from a mixed operating mechanism involving excimer formation and ionic annihilation, whose ratio varies with applied voltage and frequency. Spectroelectrochemical analysis further attributed the mixed operating mechanism to an imbalance in ion concentrations facilitated by HOMO-to-HOMO electron transfer from neutral TBRb to TpATtBu-tFFO(+) ions.

## Methods
### Calculations
Molecular structures were optimized using density functional theory (DFT) and HOMO and LUMO distributions were calculated by time-dependent DFT (TD-DFT). Both calculations were performed at the PBE0/6-31 G(d) level of theory using the polarizable continuum model in toluene within the Gaussian 16 software package[37]. A dielectric constant of 18.40 was used for the mixture of toluene and acetonitrile.

### Material characterization
[1]H and [13]C NMR spectra were obtained with JEOL ECS400. CDCl₃ was used as a deuterated solvent and all measurements were performed at ambient temperature. Chemical shifts were reported in δ (ppm), using tetramethylsilane as internal standards. Atmospheric pressure chemical ionization (APCI) mass spectra were measured with a timsTOF (Bruker).

### Preparation of solutions and films
For PL samples, 80 mM TpAT-tFFO and 80 mM TpATtBu-tFFO were each weighed into separate vials. A mixture of anhydrous toluene and anhydrous acetonitrile in 2:1 by volume was pipetted into each vial. The solutions were heated at 60 °C on a hot plate for 30 min. These 80 mM solutions were subsequently diluted to prepare concentrations down to 0.1 mM. For the hyperfluorescence PL sample, 10 mM TBRb was weighed into a vial, and then the 80 mM TpAT-tFFO solution was pipetted in. For ECL solutions, we first prepared a solution containing 10 mM TBRb and 100 mM supporting electrolyte (tetrabutylammonium hexafluorophosphate). Next, 80 mM TpAT-tFFO and 80 mM TpATtBu-tFFO were each weighed into separate vials. The TBRb solution was subsequently pipetted to each vial, followed by 30 min of heating at 60 °C.

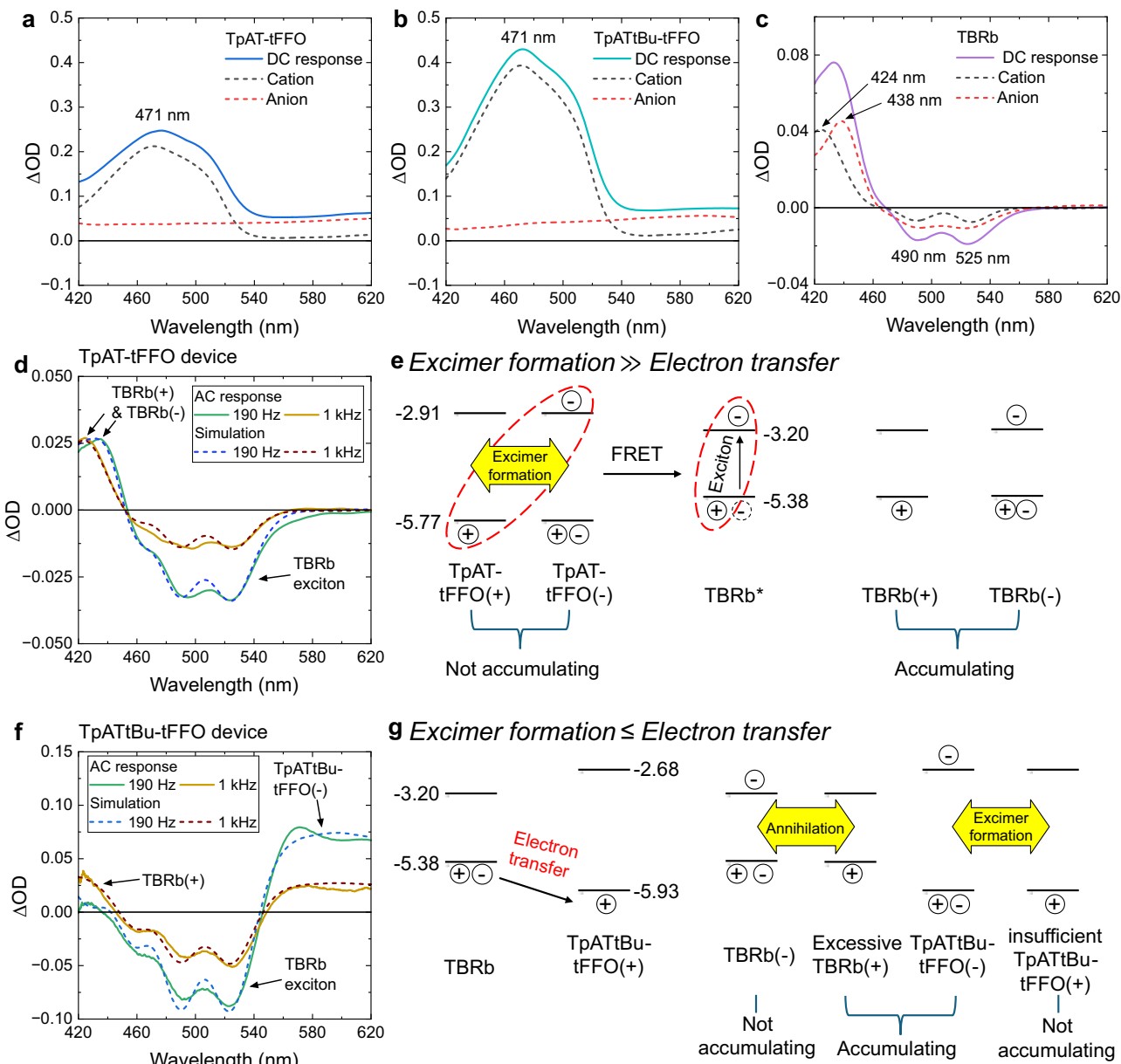

**Fig. 5 | Absorption spectroelectrochemistry.** Measured absorption spectra of **a** TpAT-tFFO, **b** TpATtBu-tFFO, and **c** TBRb ions under DC operation of devices in parallel-electrode configuration at 3.5 V, 3.5 V, and 3.0 V, respectively. The absorption bands are further resolved in coplanar-electrode configuration. **d** Spectroscopic analysis of optical density changes in TpAT-tFFO devices under AC operation at frequencies of 190 Hz and 1 kHz. The simulations use the absorption spectra of TBRb(+) and TBRb(−) ions, along with bleaching of the neutral TBRb absorption band. **e** Schematic of the ECL process based on excimer-formation on TpAT-tFFO and subsequent energy-transfer to TBRb. Due to the rapid nature of this process, TpAT-tFFO ions do not accumulate in the device. However, any TBRb ions formed by a direct redox process at the electrode will accumulate. **f** Spectroscopic analysis of optical density changes in TpATtBu-tFFO devices. **g** Schematic of the electron transfer processes, resulting in accumulation of TBRb(+) and TpATtBu-tFFO(−). The significant gap of 0.55 eV between the HOMO levels of TBRb and TpATtBu-tFFO facilitates the HOMO-to-HOMO electron transfer process.

---

For films, toluene solutions (10 mg mL⁻¹) containing 25 wt% TADF emitters and 9-(4-tert-butylphenyl)−3,6-bis(triphenylsilyl)−9H-carbazole (CzSi) were prepared, drop-cast onto quartz substrates pretreated with UV-O₃ irradiation for 30 min, and then dried under vacuum at 100 °C for 30 min.

**PL and absorption measurement**

The solutions were argon bubbled for longer than 30 min to remove residual oxygen. 700 μL quartz cuvettes were used for the PL characterization. The PL spectra and transient PL of the solutions were measured using a fluorescence lifetime spectrometer (FluoTime 250, PicoQuant) paired with a picosecond diode laser (λ = 373 nm). UV-Vis absorption for 0.1 mM and 10 mM was measured with a UV-Vis spectrometer (Cary 50, Varian). The HOMO energy levels of TpAT-tFFO and TpATtBu-tFFO were determined utilizing photoelectron yield spectroscopy (AC-3, Riken Keiki). The LUMO energy levels were calculated from the HOMO energy levels and the optical band gaps. The HOMO and LUMO levels of TBRb are taken from Ref. 38. The PLQY values were determined by using an absolute PLQY spectrometer (C9920-02, Hamamatsu Photonics).

Variable temperature photoluminescence measurements were performed on the FluoTime 250 spectrometer coupled with a cryostat (either OptistatDN, Oxford Instruments or CoolSpek UV USP-203, Unisoku). Solutions were placed in a quartz cuvette with 3 mL volume.

The sample chamber was filled with helium gas. For film measurement, we used Quantaurus-Tau Fluorescence lifetime spectrometer (C11367-01, Hamamatsu Photonics), coupled with an OptistatDN2 cryostat (Oxford Instruments).

### Photolithography and lift-off processes

A glass substrate, measuring 96 mm by 96 mm, was cleaned with acetone and isopropanol. After drying, the substrate underwent oxygen plasma treatment for 3 min. A layer of positive photoresist (AR-P 3740, Allresist GmbH) was spin-coated onto the substrate at 4000 rpm for 60 seconds, followed by a soft bake at 90 °C for 1 min. The prepared substrate was aligned and loaded into a laser writer (Picomaster 150, Raith GmbH) for pattern transfer via selective light exposure. Following exposure, the photoresist was developed (AR 300-26, Allresist GmbH, diluted 1:3 with purified water) and dried, then the substrate was placed in a vacuum chamber. A thin adhesion promoting layer of chromium and a 70-nm-thick gold layer were deposited at a base pressure of approximately $10^{-7}$ mbar. Finally, the substrate was immersed in an acetone in an ultrasonic bath for 30 min. This lift-off step removed the photoresist and any gold deposited onto the resist, leaving only the desired gold pattern on the substrate.

### Device fabrication

The fabrication of each ECLD with PE structure used a pair of rectangular glass substrates measuring 24 mm by 15 mm coated with two stripes of 2 mm-wide ITO. The substrates were cleaned using a detergent solution (2% Hellmanex III in Milli-Q water) followed by isopropyl alcohol. After drying, the ITO surface was treated with UV-ozone for 15 min. The substrate pair was bonded with NOA 68 resin (Norland Products), which was mixed with 30 μm-sized polystyrene microbeads (Sigma-Aldrich), at an angle of 90° to have four 4 mm²-size cross sections of ITO, and cured with UV light while being pressed using a custom-made holder. The edges of the bonded substrates remained open, maintaining a 30 μm gap, as the resin was applied only as droplets to the edges and corners. The bonded substrates were then transferred to a nitrogen-filled glove box, where 6.5 μL of the ECL solution was pipetted into the gap between the ITO substrates to fill the structure. The open edges were sealed with 3035BT resin (Threebond International) and cured with UV light, resulting in a production of glass-glass devices. For the glass-mirror device, after cleaning the substrate, a 120 nm-thick silver layer followed by a 50 nm-thick aluminum oxide layer was deposited onto the bare-glass side of the ITO substrates using vacuum evaporation and atomic layer deposition techniques, respectively. After coating, the substrates underwent the cleaning process again. The ITO surface was treated with UV-ozone for 3 min. A pair of silver-coated and uncoated ITO substrates was then used to fabricate each glass-mirror device following the steps described above.

For the fabrication of the ECLD in FBE configuration, the gold-patterned glass substrate was diced into nine pieces, each measuring 28 mm by 28 mm, using a diamond cutter. A diced substrate was precisely aligned with a rectangular glass substrate containing two stripes of 2 mm-wide ITO using a custom-made holder. The glasses were bonded together with NOA 68 resin containing microbeads, leaving the edges of the bonded substrates open and setting the separation between the substrates to 30 μm. The bonded substrates were transferred to the glove box, where 10 μL of the ECL solution was pipetted into the gap between the substrates. Finally, the open edges were sealed with 3035BT resin.

### Device characterization

The device was mounted on a sample holder positioned at the center of a 55 cm by 55 cm sized dark box. $JVL$ was characterized by a step-wise increase in the amplitude of sinusoidal voltage signal generated by a function generator (33220 A, Agilent Technologies). The actual $V_{rms}$ and $I_{rms}$ values were recorded by a power analyzer (GPM-8213, GW Instek) during the voltage sweep. A silicon photodiode (PDA100A2, Thorlabs) positioned 168 mm from the device measured the photocurrent in AC mode. Spectral data were collected using a fiber-coupled spectrometer (Ocean HDX, Ocean Insight). The calculation of $\Phi_{ECL}$ used the ratio of the number of outcoupled photons to the number of injected faradaic electrons. The photocurrent was converted to photon numbers with spectral calibration, assuming a Lambertian emission pattern. The number of injected Faradaic electrons was calculated using current-voltage simulations based on electrical impedance spectroscopy[29]. Operational lifetime was analyzed using the same setup, with the photocurrent recorded every second during the extended operation. The voltage amplitude was controlled in potentiostatic mode, while the amplitude was finely adjusted to ensure the rms current remained within an error range of 1.0 μA in galvanostatic mode. Custom software automized all measurements.

### Absorption spectroelectrochemistry

Spectroelectrochemical analysis was performed using a custom-built inverted microscope setup (Nikon Eclipse Ti2) equipped with 20x extra-long working distance air objective. A schematic of the measurement setup is given in Supplementary Fig. 12a. The sample was illuminated by a white LED source (pE-4000, CoolLED). A beam splitter and a digital camera (C13440 Orca-Flash 4.0, Hamamatsu) ensured precise alignment of the observation area within the active region of the device. The transmitted light spectrum was recorded using a spectrometer (Andor Shamrock 500i) with a 1-second acquisition time. First, the transmission of the white LED was measured with the device turned off ($I_{off}$ in Fig. 5a), followed by a measurement with the device turned on ($I_{on}$). Lastly, the ECL intensity was measured with the white LED switched off ($I_{ECL}$). ΔOD is calculated as $\log_{10}\{I_{off}/(I_{on}\text{-}I_{ECL})\}$. Ion absorption bands for TpAT-tFFO, TpATtBu-tFFO, and TBRb were measured using toluene:acetonitrile solutions containing 80 mM of TpAT-tFFO, 80 mM of TpATtBu-tFFO, and 10 mM of TBRb, respectively. Each solution included 100 mM of supporting electrolyte. The solution in each device had a thickness of 30 μm and devices were operated under DC voltage. Devices in parallel and coplanar electrode configuration, as shown in Supplementary Fig. 12b, were used to analyze the overall ion absorption profile and the shapes of the cation and anion absorption bands, respectively. For the simulations of the ΔOD spectra under AC driving, the ratio of each absorption band measured in DC mode was adjusted to match the overall absorption profile.

## Data availability

The primary data generated in this study have been deposited in the University of St Andrews' research information system under https://doi.org/10.17630/a26d5671-1fc8-4f51-8e9d-af46f5bd8bc1.

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

## Acknowledgements

The authors thank Matthias König for performing macro-photography of the calligraphic ECLD display. This work was financially supported by the Alexander von Humboldt Foundation (Humboldt-Professorship to M.C.G.), JSPS KAKENHI (JP20H05840, Grant-in-Aid for Transformative Research Areas, "Dynamic Exciton"; and JP23KJ1253), the JSPS Core-to-Core Program (JPJSCCA20220004), JST SPRING (JPMJSP2110 to Y.Y. and K.I.), and the International Collaborative Research Program of Institute for Chemical Research, Kyoto University (2024-126 and 2025-83). A.P. acknowledges funding from the European Molecular Biology Organization through the EMBO Postdoctoral Fellowship (675-2022). Quantum chemical calculations were performed on the SuperComputer System, Institute for Chemical Research, Kyoto University. NMR and mass-spectrometry measurements were supported by the international Joint Usage/Research Centre (iJURC) at the Institute for Chemical Research.

## Author contributions

C.-K.M., H.K., and M.C.G. conceived the project. C.-K.M. conducted PL measurements, manufactured ECLDs, and characterized them. Y.Y. and Y.K. conducted PL measurements and computations for the tFFO-based molecules. S.F. synthesized TADF emitters. A.P. contributed to the spectroelectrochemical analysis. J.F.B. built the device characterization setup. N.P. contributed to

photolithography. J.S. and O.D. performed variable-temperature PL measurements. K.I. conducted computational calculations. C.-K.M., H.K., and M.C.G. mainly wrote the manuscript.

## Funding

## Competing interests
The authors declare no competing interests.
