## [Transparent Peer Review file · Nature Communications]

Electrochemically induced hyperfluorescence based on the formation of charge-transfer excimers

Corresponding Author: Professor Malte Gather

Version 0:

Reviewer comments:

Reviewer #1

(Remarks to the Author)

Moon et al. demonstrated the application of ECLD in calligraphic displays by introducing electrochemically induced hyperfluorescence (ECiHF), which resulted in a significant enhancement of ECLD in terms of intensity, efficiency, and operational lifetime. Moreover, the exciton mechanism in the formation of ECL was scrutinized well. The innovative nature of the work and the practical results fulfill the requirements of the journal Nat. Commun. It can be accepted after the authors have resolved some fundamental issues as follows:

1. TpATtBu-tFFO is newly reported TADF molecule, the authors should add the variable temperature photoluminescence spectra and emission lifetimes to prove its TADF property.
2. The authors should supplement the photoluminescence spectra of solids to demonstrate that the post-TADF luminescence positions of TpAT-tFFO and TpATtBu-tFFO are the same as the intrinsic emission of fluorescence.
3. From Fig. 1b, the "double-decker" structures are expected to facilitate intermolecular π - π stacking at high concentrations. Consequently, to identify intramolecular as well as intermolecular CT transitions of these molecules, the differential charge density distribution should be calculated theoretically.
4. Why is the ECL efficiency of Ru(bpy)₃²⁺ missing in Table 1, and is the ECL efficiency reported in this work a relative ECL efficiency or an absolute ECL efficiency? What is the specific exciton utilization and how to calculate?
5. To further confirm the electrochemically induced hyperfluorescence, the time-resolved ECL decay curve should be measured for evaluating the lifetime of ECL signal.
6. For Fig. 5e, f, the authors should further clarify that "TpAT-tFFO device operates through a robust CT excimer mechanism, whereas the TpATtBu-tFFO device operates through a mixed mechanism involving both excimer formation and ionic annihilation". In general, there should be a significant difference in the absorption positions and intensities of the exciton complex and the ionic state.

Reviewer #2

(Remarks to the Author)

In this article, Kaji, Gather and co-workers demonstrated a significant improvement in efficiency, luminance and operation lifetime of solution-state electro-chemiluminescent device with the help of hyperfluorescence strategy, i.e. ECiHF, and showcased its use in a calligraphic display. These findings are very interesting, and the study is comprehensive with sufficient data to support the findings. Therefore, I am happy to see the publication of this manuscript in Nat. Commun. after the revision. Here are my comments:

- 1) In Figure 1c, authors demonstrated the concentration-dependent study on the photoluminescent (PL) behavior of TpAT-tFFO, which proved the transition from monomer state, then mixed state and finally to excimer state. However, the study was done in a mixed solvent of toluene and acetonitrile, while there is still a possibility that TpAT-tFFO forms aggregate at low concentrations. Authors should perform concentration-dependent PL study with pure toluene.
- 2) Follow-up question: Is there any specific reason that toluene-acetonitrile is chosen/used in this ECiHF system? Authors should try other solvent combinations, e.g. toluene-hexane, THF-MeCN?
- 3) More discussions on the photophysical behavior of TpAT-tFFO and TpATtBu-tFFO should be put into the main manuscript. Comparing supplementary tables S1 and S2, why is the decay lifetime of TpATtBu-tFFO much shorter than that of TpAT-tFFO in monomer state? I believe this is due to the higher non-radiative decay process of TpATtBu-tFFO, showing a lower PLQY. If so, the calculation of kRISC will be over-estimated, which is in the order of 10⁸ s⁻¹.
- 4) Authors should explain why the PLQY of TpAT-tFFO decreased a lot when the excimer formed, but TpATtBu-tFFO

excimer showed an insignificant drop in PLQY.

5) The temperature-dependent PL study should be performed on monomer and excimer state to confirm the TADF properties.

6) In ECIHF device, the ratio of TBRb to sensitizer is 1:8, which is ~ 10 mol%. How about the other ratio? In HF-OLEDs, a slightly change in the doping concentration of terminal emitter always results in different device performance.

7) In Figure 2b, the emission spectrum of glass-mirror device is narrower than that of glass-glass device. Authors should explain this observation.

8) In HF-OLEDs, beyond FRET from high-energy sensitizer to low-energy terminal emitter, there is a possibility to up-convert/blue-shift the emission (see. ref. Nat. Photonics 2024, 18, 554; Nat. Photonics 2024, 18, 516 and Nat. Photonics 2021, 15, 203). I wonder if it is possible in ECIHF devices.

9) For ECL solution, a supporting electrolyte is added to enhance the conductivity of the device. Supporting electrolyte always affects the redox couples in cyclic voltammetry measurement. What is the concentration effect of this electrolyte on the ECL performance? And how about the counter anion effect, e.g. tetrabutylammonium perchlorate instead of tetrabutylammonium hexafluorophosphate? What is the resistance of the solution of the ECL device?

Reviewer #3

(Remarks to the Author)

This paper reports the fundamental device properties of solution-processed electroluminescent devices (ECLDs) based on a newly proposed architecture. The authors achieved notable improvements in device performance, including luminance, efficiency, and operational stability. They also investigated the excimer characteristics of TADF molecules through optical spectroscopy. The use of a hyperfluorescence mechanism in this device architecture is particularly interesting.

Although the devices are not yet ready for practical applications, the improvements reported are significant and relevant to the development of other types of electroluminescent devices. From this perspective, I judge that this paper merits publication in Nature Communications. However, the following points should be addressed in the revised version.

The spectroscopic results shown in Figure 5 are intriguing. However, the interpretation of the data in Figures 5e and 5f is somewhat confusing. The authors attribute the observed signals to ions of TBRb or of the TADF molecule. This assignment is questionable, as the expected absorption features from TADF molecule ions would likely include a strong positive signal in the 460–520 nm region, which appears to be absent.

In Figure 5d, the ionized TBRb shows both a positive absorption peak at 434 nm and negative bleaching signals at 490 nm and 525 nm. In contrast, Figure 5e shows that the positive signal near 430 nm is significantly weaker than in the TBRb ion spectrum, while the bleaching signals at 490 nm and 525 nm remain similar in intensity. This suggests that the bleaching cannot be attributed solely to the ionized species.

Instead, the bleaching might partly originate from luminescent excitons of TBRb, which is plausible given TBRb's long-lived excited state, as shown in Figure 1h. Notably, the intensity difference in the positive signal between the 190Hz and 1kHz conditions may reflect a difference in emission intensity. Indeed, Figure 5e suggests stronger emission at 190 Hz than at 1 kHz, consistent with the left panel of Figure 4b (despite the difference in applied voltage).

However, the data in Figure 5f does not align with this interpretation. In particular, the pronounced signal around 580 nm suggests the involvement of an additional mechanism. While this is just one possible interpretation, the current explanation remains ambiguous. A clearer and more consistent explanation should be included in the revised manuscript.

Version 1:

Reviewer comments:

Reviewer #1

(Remarks to the Author)

Although the time-resolved ECL decay curve was not measured for evaluating the lifetime of ECL signal, the authors have addressed most of questions. Now it is can be published as it is.

Reviewer #2

(Remarks to the Author)

Authors have already revised the manuscript according to my comments, so I recommend the acceptance of this manuscript for publication as is.

Reviewer #3

(Remarks to the Author)

Overall, the authors have addressed my concerns by providing additional experimental results, which I believe will be relevant to the readers of Nature Communications. Therefore, I recommend this manuscript for acceptance for publication in Nature Communications.

Reviewer #1

Moon et al. demonstrated the application of ECLD in calligraphic displays by introducing electrochemically induced hyperfluorescence (ECiHF), which resulted in a significant enhancement of ECLD in terms of intensity, efficiency, and operational lifetime. Moreover, the exciton mechanism in the formation of ECL was scrutinized well. The innovative nature of the work and the practical results fulfill the requirements of the journal Nat. Commun. It can be accepted after the authors have resolved some fundamental issues as follows:

We thank the reviewer for this positive overall assessment of our work and are happy to address the helpful comments raised by the reviewer point-by-point.

1. *TpATtBu-tFFO is newly reported TADF molecule, the authors should add the variable temperature photoluminescence spectra and emission lifetimes to prove its TADF property.*

We have now performed PL measurements of TpATtBu-tFFO for varying temperatures to analyze the intersystem crossing rate, k_{ISC} , the reverse intersystem crossing rate, k_{RISC} , and the activation energy for TADF. These results have been added to the revised manuscript as follows:

In Supplementary Information,

“

Supplementary Fig. 7. Temperature-dependent PL of a thin film of 25 wt% TpATtBu-tFFO in CzSi. **a** Time-resolved PL decay measured across temperatures from 200 K to 300K. **b** Temperature dependence of the rates of intersystem crossing (k_{ISC}) and reverse intersystem crossing (k_{RISC}), along with the activation energy as determined from Arrhenius fits to the data. At room temperature, the k_{ISC} and k_{RISC} rates were $4.2 \times 10^6 \text{ s}^{-1}$ and $1.4 \times 10^5 \text{ s}^{-1}$, respectively.

”

2. *The authors should supplement the photoluminescence spectra of solids to demonstrate that the post-TADF luminescence positions of TpAT-tFFO and TpATtBu-tFFO are the same as the intrinsic emission of fluorescence.*

We have performed time-resolved measurements for PL spectra of thin films of TpAT-tFFO and TpATtBu-tFFO doped in CzSi. The spectra of prompt and delayed fluorescence peak at

similar wavelengths, with the observed red shift post excitation attributed to the solid-state solvation effects on the charge-transfer states (Northey, *J. Mater. Chem. C*, 2017, 42, 11001). These observations confirm that the delayed fluorescence originates from the same singlet excited state as the prompt fluorescence.

We have now included this measurement in Supplementary Information as follows:

“

Supplementary Fig. 8. Temporal evolution of thin-film photoluminescence spectrum post excitation. Measurements for **a** TpAT-tFFO and **b** TpATtBu-tFFO doped in CzSi films show that both prompt and delayed fluorescence originate from the same singlet excited state. The observed red shift is attributed to solid-state solvation effects on the charge-transfer states.

”

We have also added the following text to the main manuscript, incorporating answers to the reviewer’s first and second questions,

Line 246 on page 12,

“We also studied ECiHF operation using the second excimer-forming material, TpATtBu-tFFO, which has a higher band gap (3.25 eV) than TpAT-tFFO (2.86 eV) and also features TADF in thin films (see Supplementary Figs. 7 and 8).”

The Method section has been supplemented as follows:

“Preparation of solutions and films

...

For films, toluene solutions (10 mg/mL) containing 25 wt% TADF emitters and 9-(4-tert-butylphenyl)-3,6-bis(triphenylsilyl)-9H-carbazole (CzSi) were prepared, drop-cast onto quartz substrates pretreated with UV-O₃ irradiation for 30 minutes, and then dried under vacuum at 100 °C for 30 minutes”

“PL and absorption measurement

...

For film measurement, we used Quantaaurus-Tau Fluorescence lifetime spectrometer (C11367-01, Hamamatsu Photonics), coupled with an OptistatDN2 cryostat.”

3. From Fig. 1b, the “double-decker” structures are expected to facilitate intermolecular π - π stacking at high concentrations. Consequently, to identify intramolecular as well as intermolecular CT transitions of these molecules, the differential charge density distribution should be calculated theoretically.

We simulated the excimer formation in a 2:1 toluene:acetonitrile solution containing 80 mM of TpAT-tFFO by performing 50 MD simulations. The system included 8, 627, and 639 molecules of TpAT-tFFO, toluene, and acetonitrile, respectively, to reflect their molecular ratios in the device; half of the TpAT-tFFO molecules were assigned as cations and the other half as anions. The simulation resulted in three cation–anion aggregated pairs of TpAT-tFFO. We then analyzed the excited state properties of these pairs through TD-DFT calculations. In all cases, the excimer state (intermolecular CT transition) showed both a lower S_1 energy and a smaller ΔE_{ST} than the monomer excited state (intramolecular CT transition).

We have included these new results in Supplemental Information as follows:

“

Supplementary Table 1. Calculated S_1 energies, $S_1(\text{monomer})-S_1(\text{agg. pair})$ energy differences, T_1 energies and ΔE_{ST} of the cation–anion aggregated pairs of TpAT-tFFO shown in Supplementary Figure 1, where agg. pair denotes an aggregated pair.

	S_1 / eV	$S_1(\text{monomer})-S_1(\text{agg. pair}) / \text{eV}$	T_1 / eV	$\Delta E_{ST} / \text{meV}$
Monomer	2.849	-	2.837	12
Agg. pair #1	2.381	0.468	2.373	8
Agg. pair #2	2.328	0.521	2.321	7
Agg. pair #3	2.490	0.359	2.487	3

“

“

Supplementary Fig. 1 Differential charge density distributions ($\Delta\rho$) of the three cation–anion pairs of TpAT-tFFO for the $S_0 \rightarrow S_1$ transition. In all molecular pairs, the left side represents the cation and the right side represents the anion. Blue and purple indicate negative and positive $\Delta\rho$, respectively. The monomer structures were optimized using PBE0/6-31G(d) level of theory using the polarizable continuum model in toluene. A dielectric constant of 18.40 was used for the mixture of toluene and acetonitrile. MD simulations were performed on 50 systems to reproduce high concentration conditions (80 mM of TpAT-tFFO in toluene/acetonitrile (2:1 vol%). The molecular ratio of TpAT-tFFO:toluene:acetonitrile was set to 8:627:639 molecules with half of the TpAT-tFFO molecules assigned as cations and the other half as anions. The MD simulations were performed in the NVT ensemble for 1.0 ns at 300 K with a box length of 5.5 nm. The S_1 and T_1 energies and the differential charge density distributions ($\Delta\rho$'s) were calculated for three cation-anion aggregated pairs of TpAT-tFFO using the TD-PBE0/6-31G(d) method using the polarizable continuum model in toluene and a dielectric constant of 18.40. Molecular structure optimizations and TD-DFT calculations were performed by the Gaussian 16 program package². The initial arrangements of MD simulations were performed by the Packmol package³. The MD simulations were performed by the LAMMPS GPU package⁴. The calculations of $\Delta\rho$'s were performed by the Multiwfn package⁵.

”

And in the Reference section of Supplementary Information,

“

2. Frisch, M. J. *et al.* Gaussian 16, Revision C.01, Gaussian, Inc., Wallingford CT, 2016.
3. Martínez, L., Andrade, R., Birgin, E. G., Martínez, J. M. Packmol: A package for building initial configurations for molecular dynamics simulations. *J. Comput. Chem.* **30**, 13, 2157-2164 (2009).
4. Thompson, A. P. *et al.* LAMMPS—a flexible simulation tool for particle-based materials modeling at the atomic, meso, and continuum scales. *Comput. Phys. Commun.* , **271**, 108171 (2022).
5. Lu, T. A comprehensive electron wavefunction analysis toolbox for chemists, Multiwfn, *J. Chem. Phys.* **161**, 082503 (2024).

”

We have also added the following text to the main manuscript:

Line 123 on page 6,

“Our simulation model, using molecular dynamics simulations and time-dependent density functional theory calculations, also yielded cation-anion aggregated pairs of TpAT-tFFO with lower singlet-state energies in the intermolecular CT transition than in the intramolecular CT transition in a toluene:acetonitrile mixture (see Supplementary Table 1 and Supplementary Fig. 1).”

4. Why is the ECL efficiency of $\text{Ru}(\text{bpy})_3^{2+}$ missing in Table 1, and is the ECL efficiency

reported in this work a relative ECL efficiency or an absolute ECL efficiency? What is the specific exciton utilization and how to calculate?

The ECL efficiencies listed in Table 1 are absolute values, calculated as the ratio of the number of outcoupled photons to the number of injected faradaic electrons. As detailed in the Device characterization subsection in Methods, we measured optical flux using a calibrated photodiode positioned 168 mm away from the device and converted the flux into photon numbers by using measured spectral data. The number of injected Faradaic electrons were calculated using current-voltage simulations based on electrical impedance spectroscopy, as detailed in our previous publication, Reference 29 in our revised manuscript.

We have added further details to clarify the efficiency parameter and our methodology as follows:

Line 177 on page 8,

“The **absolute** ECL quantum efficiency (Φ_{ECL}) and luminous efficacy (LE) shown reached maximum values of 0.56% and 1.66 lm/W, respectively, ...”

In Methods,

“**Device characterization**”

The device was mounted on a sample holder positioned at the center of a 55 cm by 55 cm sized dark box. *JVL* was characterized by a step-wise increase in the amplitude of sinusoidal voltage signal generated by a function generator (33220A, Agilent Technologies). The actual V_{rms} and I_{rms} values were recorded by a power analyzer (GPM-8213, GW Instek) during the voltage sweep. A silicon photodiode (PDA100A2, Thorlabs) positioned 168 mm from the device measured the photocurrent in AC mode. Spectral data were collected using a fiber-coupled spectrometer (Ocean HDX, Ocean Insight). ~~A Lambertian distribution was assumed in the calculation of Φ_{ECL} .~~ **The calculation of Φ_{ECL} used the ratio of the number of outcoupled photons to the number of injected faradaic electrons. The photocurrent was converted to photon numbers with spectral calibration, assuming a Lambertian emission pattern. The number of injected Faradaic electrons was calculated using current-voltage simulations based on electrical impedance spectroscopy²⁹.** Operational lifetime was analyzed using the same setup, with the photocurrent recorded every second during the extended operation. The voltage amplitude was controlled in potentiostatic mode, while the amplitude was finely adjusted to ensure the rms current remained within an error range of 1.0 μA in galvanostatic mode. Custom software automated all measurements.”

Additionally, we have newly added the most recently estimated absolute ECL efficiency value for Ru(bpy)₃²⁺ in Table 1 (Φ_{ECL} =0.08%, Kang, *Advanced Functional Materials*, 2025, 35, 2417514). Please note that this table only compares AC-driven ECLDs in parallel ITO electrode configurations. Thus, studies using glass tube cells with three-electrode or rotating ring-disk electrode (RRDE) configurations are not included in the table. In RRDE setups, the absolute ECL efficiency of the Ru(bipy)₃²⁺ was reported to be 6% (Bard, *JACS*, 1973, 95, 6582). In three-electrode configurations, the absolute ECL efficiency of Ru(bpy)₃²⁺ for the annihilation mechanism was estimated to be 0.0019% by triangle-voltage operation and 2.43% by pulsed voltage operation (Ding, *Anal. Chem.* 2021, 93, 11626). In a co-reactant mechanism involving TPrA, the same authors measured at 3.1% and 10%, respectively, for the two configurations

(Ding, *J. Phys. Chem. C* 2021, 125, 22274). However, we did not include these data in the table as a direct comparison between devices in parallel-electrode and glass-tube configuration would likely cause confusion due to a number of fundamental differences, including in device operation (DC, AC, pulse, etc.), measurement methods (e.g., actinometer, photodiode, integrating sphere, etc.), and calculation methods (e.g., whether to include self-absorption, electrode reflection, etc.).

We have updated Table 1 and added a further citation as follows:

“

Table 1. Summary of characteristics of ECLDs with PE configuration using ITO electrodes, AC driving, and different operating mechanisms. *M* indicates a device with a mirror coating on one substrate side. *T* indicates a device with a mesoporous TiO₂ electrode. Reference ²⁹ reports Φ_{ECL} values of 0.35% and 0.52% for operation at 300 Hz and 100 Hz, respectively; this table shows the Φ_{ECL} value at 300 Hz, at which the excimer devices showed higher luminance. Reference ¹⁸ measured the lifetimes at an initial luminance of approximately 115 cd/m² and demonstrated a further improvement in lifetime for a floating bipolar electrode configuration. Reference ³⁴ measured the lifetime at an initial luminance of 165 cd/m².

Operating mechanism	Material	Medium	L_{max} (cd/m ²)	LE_{max} (lm/W)	Φ_{ECL} (%)	LT_{50} (sec), $L_0 \sim 100$ cd/m ²
CT excimer (this work)	TBRb & TpAT-tFFO	toluene- acetonitrile	3650, 6220 (M)	1.66, 2.66 (M)	0.56, 0.93 (M)	1252
Exciplex ^{28,29}	TBRb, TAPC & TPBi	toluene- acetonitrile	1250, 2260 (T)	1.17, 2.06 (T)	0.38	75
Annihilation ²⁸	TBRb	toluene- acetonitrile	390	0.74	-	26
Annihilation ^{18,35,36}	Ru(bpy) ₃ ²⁺	ionic liquid	700	0.20	0.08	300
Annihilation ³⁴	Ru(bpy) ₃ ²⁺ & TiO ₂ nanoparticles	propylene carbonate	165	-	-	1000

”

And in the Reference section,

“35. Yee, H. *et al.* Transparent Electrode-Free Light-Emitting Devices Exploiting Ion Transport-Controlled Electrochemiluminescence. *Adv. Funct. Mater.* 35, 2417514 (2025).”

We also have added comments on strategies to extend the operating lifetime of ECLD from the newly cited paper by innovative electrode structures as follows:

Line 213 on page 10,

“Innovations in electrode structures have the potential to increase the operational lifetime of ECLDs operating through ECiHF even further. For example, Kang *et al.* utilized gold electrodes for ECLDs and employed Ru(bpy)₃²⁺ luminophores for annihilation. These strategies effectively extended the LT_{50} lifetime values to 81 minutes and 600 minutes in floating bipolar¹⁸ and coplanar electrode arrangements³⁵, respectively.”

5. To further confirm the electrochemically induced hyperfluorescence, the time-resolved ECL decay curve should be measured for evaluating the lifetime of ECL signal.

Showing hyperfluorescence in time-resolved ECL data would indeed provide strong additional evidence of ECiHF. However, unfortunately, the ECL signal generally persists for a few milliseconds after ECLDs have been switched off (Ishimatsu, *Angew. Chem.* 2014, 126, 7113–7116; Yang, *SmartMat.* 2023, 4, e1149). While the explanations given in the literature for this after-glow vary, a plausible reason is that it is due to the presence of diffusing molecular ions that continue to sustain the ECL process for a while even after the device is switched off. The lifetime of delayed fluorescence of TBRb:TpAT-tFFO was measured at 172 ns in photoluminescence, orders of magnitude shorter than this afterglow, which makes it practically impossible to resolve in the presence of this ion diffusion.

Instead, we therefore had to rely on more indirect measurements and concluded that the hyperfluorescence observed in photoluminescence (Fig. 1h), the excimer emission seen in the absence of a terminal emitter in an ECLD (Supplementary Fig. 3), and the significant increases in luminance and ECL intensity upon addition of TpAT-tFFO (Table 1) together strongly support the dominance of the proposed ECiHF process.

6. For Fig. 5e, f, the authors should further clarify that “TpAT-tFFO device operates through a robust CT excimer mechanism, whereas the TpATtBu-tFFO device operates through a mixed mechanism involving both excimer formation and ionic annihilation”. In general, there should be a significant difference in the absorption positions and intensities of the exciton complex and the ionic state.

This is an important point, and we have now performed additional spectroscopic experiments and analysis to elucidate this question. Our extensive answer to the question about spectroelectrochemistry raised by Reviewer #3 provides a thorough answer to this point, and we kindly refer the reviewer to this answer.

In short, we have newly added data that now separately resolves the absorption bands of cations and anions of three involved materials—TpAT-tFFO, TpATtBu-tFFO, and TBRb. This was achieved by using devices in coplanar electrode configurations and performing simulations of the Δ OD curves in AC operation based on this new data. This analysis confirmed again that the TpAT-tFFO device operates through a robust CT excimer mechanism, whereas operation of the TpATtBu-tFFO device involves mixed exciplex formation and annihilation processes. The Δ OD simulations indicate the presence of a rapid HOMO-to-HOMO electron transfer process from TpATtBu-tFFO(+) ions to TBRb molecules, which leads to exciton quenching and accelerates degradation.

Reviewer #2

In this article, Kaji, Gather and co-workers demonstrated a significant improvement in efficiency, luminance and operation lifetime of solution-state electro-chemiluminescent device with the help of hyperfluorescence strategy, i.e. ECiHF, and showcased its use in a calligraphic display. These findings are very interesting, and the study is comprehensive with sufficient data to support the findings. Therefore, I am happy to see the publication of this manuscript in Nat. Commun. after the revision.

We sincerely appreciate the reviewer's positive comments and their very helpful questions regarding ECiHF. We have thoroughly addressed each question below and endeavored to resolve all concerns.

However, due to resource limitations, we kindly request the reviewer's understanding that we were not able to carry out some of the supplementary experiments requested. Currently, TpAT-tFFO and TpATtBu-tFFO are not commercially available and the amount we have been able to synthesize is limited. We decided that it would be most important to conduct experiments related to questions 1 (PL measurements in toluene), 3 (revalidation of k_{RISC} and k_{ISC} for TpATtBu-tFFO), and 5 (temperature-dependent PL experiments). On the other hand, questions requiring extensive device optimization, i.e., questions 2 (solvent mixture optimization), 6 (optimization of excimer-to-emitter material ratio), and 9 (electrolyte material optimization), could not be explored with the current material set. Instead, for questions 6 and 9, we provide indirect evidence based on additional experiments that we performed with a previously reported material set that uses a different operation mechanism, specifically the TAPC:TPBi exciplex system (Gather, *Advanced Materials*, 2023, 35, 2302544). We have also added a discussion about possible solvent combinations for ECL to address question 2.

1. In Figure 1c, authors demonstrated the concentration-dependent study on the photoluminescent (PL) behavior of TpAT-tFFO, which proved the transition from monomer state, then mixed state and finally to excimer state. However, the study was done in a mixed solvent of toluene and acetonitrile, while there is still a possibility that TpAT-tFFO forms aggregate at low concentrations. Authors should perform concentration-dependent PL study with pure toluene

Acetonitrile is indeed a highly polar solvent with very low solubility for both TpAT-tFFO and TpATtBu-tFFO, and thus one might expect that it induces aggregation of these molecules when mixed into toluene. As the reviewer suggested, we have now measured PL spectra of TpAT-tFFO and TpATtBu-tFFO at concentrations of 0.1 mM and 10 mM, respectively, each in toluene. As shown in Figure R1 (reviewer only) below, we observed redshifts of approximately 8 nm and 7 nm for TpAT-tFFO and TpATtBu-tFFO, respectively, when going from diluted to concentrated solutions. These shifts are significantly smaller than the 48 nm and 40 nm redshifts observed for the mixed toluene-acetonitrile solutions.

Figure R1. PL measurement in pure toluene. **a**, TpAT-tFFO at 0.1 mM and 80 mM. **b**, TpATtBu-tFFO at 0.1 mM and 80 mM.

This observation suggests that mixing acetonitrile with toluene is indeed likely to induce aggregation when the molecules are excited in concentrated solutions. This finding is similar to the principle of aggregation-induced electrochemiluminescence (AIECL, Carrara, *et al.*, *JACS*. 2017, 139, 14605-14610; Wei, *et al.*, *Chem. Eur. J.* 2019,25, 12671-12683) that occur when using highly polar solvents such as water. However, while AIECL features the formation of supramolecules in the ground state, leading to the appearance of new absorption bands, such changes were not observed in toluene-acetonitrile mixed solutions of TpAT-tFFO and TpATtBu-tFFO (as shown in revised Figure 1e and revised Supplementary Figure 11d). Therefore, we regard this emission shift as aggregation upon excitation, i.e., excimer formation.

To reflect our new findings, we have revised the manuscript as follows:

Figure 1e,

“

Original figure

e, Molar attenuation of TpAT-tFFO measured at concentrations of 0.1 mM and 10 mM.

Revised figure

e, Molar attenuation of TpAT-tFFO at concentrations of 0.1 mM and 10 mM, measured in a toluene:acetonitrile mixture and pure toluene.

”

Line 128 on page 6,

“Unlike the emission, the absorption spectra (spectrally resolved molar extinction coefficient) of TpAT-tFFO at concentrations of 0.1 mM and 10 mM are largely identical, with no red shift of the CT absorption observed with increasing concentration (Fig. 1e). This indicates negligible intermolecular interactions of TpAT-tFFO in the ground state, even though strong intermolecular binding occurs upon excitation.

Fig. 1e shows the absorption (spectrally resolved molar extinction coefficient) and emission spectra of TpAT-tFFO at concentrations of 0.1 mM and 10 mM, measured in a toluene:acetonitrile mixture and in pure toluene, respectively. The red-shift in emission spectrum is significant in toluene:acetonitrile but the shift is only 7 nm in pure toluene, indicating that the aggregation is facilitated by adding acetonitrile, a high polarity solvent. The aggregation in the ground state is negligible, given that the absorption spectrum of TpAT-tFFO remained nearly unchanged regardless of mixing solvent or increasing concentration.”

The corresponding figure showing the absorption spectra of TpATtBu-tFFO in Supplementary Information (originally Supplementary Fig. 5b, now Supplementary Fig. 11d) has also been revised as follows:

“

b, molar attenuation coefficient.

d, Molar attenuation of TpATtBu-tFFO at concentrations of 0.1 mM and 10 mM, measured in a toluene:acetonitrile mixture and pure toluene.

”

2. Follow-up question: Is there any specific reason that toluene-acetonitrile is chosen/used in this ECiHF system? Authors should try other solvent combinations, e.g. toluene-hexane, THF-MeCN?

Mixing a polar solvent with a less polar solvent is a common strategy to enhance ECL performance (Phighin, *J. Electrochem. Soc.* 1975, 122, 619, Nishimura, *Jpn. J. Appl. Phys.* 2001, 40, L1323-L1326). Less polar solvents, such as toluene, dissolve neutral molecules well, while their corresponding oxidized/reduced forms and the electrolytes required for ECLD

operation are more soluble in polar solvents like acetonitrile. A mixture of two solvents leads to a co-operative effect and thus enhances ECL.

When choosing solvent combinations, we considered three main criteria: (1) high solubility of organic semiconductor molecules in the less polar solvent, (2) good miscibility between the solvents, and (3) a wide electrochemical window. The combinations of toluene-hexane and the THF-acetonitrile suggested by the reviewer meet conditions 2 and 3, and we believe they are all likely to meet the high solubility requirement as well.

In the future, it might therefore be promising to explore these and other solvent combinations. Furthermore, it will also be interesting to explore the effect of solvents on spin upconversion (Ishimatsu, *Chem Eur J.* 2016, 22, 4889) of excimers. However, we request the reviewer's understanding that producing a comprehensive ECLD dataset for different solvent combinations is beyond the scope of the present study and in fact not feasible due to resource limitations.

3. More discussions on the photophysical behavior of TpAT-tFFO and TpATtBu-tFFO should be put into the main manuscript. Comparing supplementary tables S1 and S2, why is the decay lifetime of TpATtBu-tFFO much shorter than that of TpAT-tFFO in monomer state? I believe this is due to the higher non-radiative decay process of TpATtBu-tFFO, showing a lower PLQY. If so, the calculation of k_{RISC} will be over-estimated, which is in the order of 10^8 s^{-1} .

We thank the reviewer for this remark, which triggered us to thoroughly re-evaluate our transient PL measurements.

For the PL measurement with solution samples, we used a 1 mm path-length quartz cuvette, and a pulsed laser diode with a wavelength of 373 nm to excite the solution. Figure R3a shows the raw transient PL signal of TpATtBu-tFFO solution (black line), which is well described by a biexponential decay function with fluorescence lifetimes of $t_1 = \sim 4 \text{ ns}$ and $t_2 = \sim 200 \text{ ns}$. In the original manuscript, we regarded these as prompt and delayed fluorescence and thus used these parameters to calculate both k_{ISC} and k_{RISC} , obtaining values on the order of 10^8 s^{-1} for both.

However, we have now found that this original analysis was unfortunately incorrect. TpATtBu-tFFO has a molar attenuation coefficient of only $66.2 \text{ M}^{-1}\text{cm}^{-1}$ at the used excitation wavelength of 373 nm. Combined with the short path-length of the cuvette, this renders the signal extremely sensitive to unspecific emission (fluorescence from trace contaminants in solvent or cuvette glass) and scattering of excitation light not blocked by the detection monochromator. The red line in Figure R2a (reviewer-only figure) represents a measurement with a blank cuvette containing only the solvent, showing a fast PL decay with a lifetime of approximately 4 ns. The fast decay corresponds exactly to the fast decay previously observed for the TpATtBu-tFFO transient PL signal. The other investigated material, TpAT-tFFO, has a much higher attenuation coefficient of $390 \text{ M}^{-1}\text{cm}^{-1}$, and therefore transient PL measurements on this material were not influenced by this effect.

In this revision, we remeasured transient PL for TpATtBu-tFFO over an extended time scale, added a suitable long-pass filter to more effectively block the excitation laser light and performed a calibration with a blank sample signal. This provided us with properly resolved prompt and delayed fluorescence with lifetimes of 212 ns and 2.17 μs , respectively, as shown in Figure R2b.

Figure R2. Transient PL measurement of TpATtBu-tFFO solution. a, Measurements of TpATtBu-tFFO solution sample and blank cuvette sample. b, Measurement up to 10 microseconds post-excitation.

With our improved measurement routine, we have re-measured the time-resolved PL across a concentration range between 0.1 mM and 80 mM. This resulted in k_{ISC} values on the order of 10^6 s^{-1} and k_{RISC} values in the order of 10^5 s^{-1} . We have also performed the variable-temperature PL measurement for a 25 wt% TpATtBu-tFFO doped in CzSi film, showing k_{ISC} and k_{RISC} values of $4.2 \times 10^6 \text{ s}^{-1}$ and $1.4 \times 10^5 \text{ s}^{-1}$, respectively, similar to the values measured for TpATtBu-tFFO solutions.

We again thank the reviewer for this very helpful comment and apologize for not having spotted this mistake before.

We have replaced Supplementary Figures 11b, 11c (originally Supplementary Figures 5c, 5d) and Supplementary Table 3 (originally Supplementary Table 2) with the new data, and have newly included the film analysis in Supplementary Fig.7 as follows:

Supplementary Figs. 11b and 11c

“

Original Supplementary Figure 5c

Revised Supplementary Figure 10b

Original Supplementary Figure 5d

Revised Supplementary Figure 10c

c, transient PL, and **d**, PLQY ratio of delayed fluorescence to prompt fluorescence of TpATtBu-tFFO for various concentrations in mixed toluene-acetonitrile solutions (2:1 by volume).

b, transient PL, and **c**, PLQY ratio of delayed fluorescence to prompt fluorescence of TpATtBu-tFFO for various concentrations in mixed toluene-acetonitrile solutions (2:1 by volume).

”

Supplementary Table 3 (originally Supplementary Table 2)

“

Supplementary Table 3. TADF parameters of TpATtBu-tFFO at various concentrations in mixed toluene and acetonitrile solutions. The lifetime parameters are extracted from transient PL curves using the equation, $I(t) = I_p \exp(-t/\tau_p) + I_d \exp(-t/\tau_d)$. ~~We found that TpATtBu-tFFO is a TADF molecule with particularly high sensitivity to oxygen. Prior to the PL measurement, each solution was purged with argon for 30 minutes to minimize oxygen interference. However, trace oxygen may still be present in the solution and quench triplet emission from TpATtBu-tFFO during the measurement, potential leading to errors in the ISC and RISC parameters of TpATtBu-tFFO reported here.~~ **The analysis of k_{ISC} and k_{RISC} was performed using the method described in Methods section of reference 1.**

Concentration (mM)	State	I_p	τ_p (ns)	I_d	τ_d (μ s)	Φ_d/Φ_p	Φ_{PL}	k_{ISC} ($10^6/s$)	k_{RISC} ($10^5/s$)	$k_{ISC}k_{RISC}$ ($10^{12}/s$)
0.1	Monomer	0.963	142	0.037	1.92	0.51	0.52	2.27	8.24	1.88
0.3	Monomer	0.958	161	0.042	1.95	0.53	0.52	2.03	8.26	1.68
1	Mixed	0.938	190	0.061	1.98	0.68	-	1.97	9.24	1.82
3	Mixed	0.932	212	0.068	2.17	0.74	-	1.83	8.78	1.61
10	Excimer	0.942	198	0.058	2.43	0.76	0.50	2.02	7.78	1.57
30	Excimer	0.935	239	0.065	3.03	0.88	0.50	1.80	6.76	1.21
80	Excimer	0.921	265	0.079	3.13	1.01	0.50	1.70	7.19	1.22

”

Supplementary Figure 7,

“

Supplementary Fig. 7. Temperature-dependent PL of a thin film of 25 wt% TpATtBu-tFFO in CzSi. **a** Time-resolved PL decay measured across temperatures from 200 K to 300K. **b** Temperature dependence of the rates of intersystem crossing (k_{ISC}) and reverse intersystem crossing (k_{RISC}), along with the activation energy as determined from Arrhenius fits to the data. At room temperature, the k_{ISC} and k_{RISC} rates were $4.2 \times 10^6 \text{ s}^{-1}$ and $1.4 \times 10^5 \text{ s}^{-1}$, respectively.

”

In the main text, we have deleted the description of TpATtBu-tFFO excimer emission “exhibiting strong delayed fluorescence” and “potentially offering improved ECiHF performance” to align with new measurement data. We have revised the text to include the newly measured RISC rate of TpATtBu-tFFO as follows:

Line 252, page 12,

~~“The excimer emission also exhibited strong delayed fluorescence, with higher Φ_d/Φ_p ratio than the monomer emission (see Supplementary Table 2 and Supplementary Fig. 5). The TpATtBu-tFFO excimer emission spectrum showed a stronger overlap with TBRb absorption than the TpAT-tFFO excimer (see Supplementary Fig. 10), thus potentially offering more efficient FRET and improved ECiHF performance.~~ but a slower RISC rate of $7.2 \times 10^5 \text{ s}^{-1}$ (see Supplementary Table 3 and Supplementary Fig. 11).

The Method section has been supplemented as follows:

“Preparation of solutions and films

...

For films, toluene solutions (10 mg/mL) containing 25 wt% TADF emitters and 9-(4-tert-butylphenyl)-3,6-bis(triphenylsilyl)-9H-carbazole (CzSi) were prepared, drop-cast onto quartz substrates pretreated with UV-O₃ irradiation for 30 minutes, and then dried under vacuum at 100 °C for 30 minutes”

“PL and absorption measurement

...

For film measurement, we used Quantaaurus-Tau Fluorescence lifetime spectrometer (C11367-01, Hamamatsu Photonics), coupled with an OptistatDN2 cryostat”

Apart from this issue, we have noticed that the analysis of k_{ISC} and k_{RISC} for 1 mM, 3 mM, and 10 mM TpAT-tFFO was omitted in Supplemental Table 2 (originally Supplementary Table 1). We have now included these parameters as follows:

Supplementary Table 2 (originally Supplementary Table 1),

“

Supplementary Table 2. TADF parameters of TpAT-tFFO at various concentrations in mixed toluene and acetonitrile solutions. The lifetime parameters are extracted from transient PL curves using the equation, $I(t) = I_p \exp(-t/\tau_p) + I_d \exp(-t/\tau_d)$. The analysis of k_{ISC} and k_{RISC} used was performed using the method described in Methods section of reference¹.

Concentration (mM)	State	I_p	τ_p (ns)	I_d	τ_d (ns)	Φ_d/Φ_p	Φ_{PL}	k_{ISC} (10 ⁷ /s)	k_{RISC} (10 ⁷ /s)	$k_{\text{ISC}}k_{\text{RISC}}$ (10 ¹⁴ /s)
0.1	Monomer	0.837	17.3	0.144	977	9.73	0.84	4.02	1.42	5.70
0.3	Monomer	0.883	18.3	0.128	1150	9.16	0.84	4.03	1.07	4.31
1	Mixed	0.837	20.2	0.158	928	8.65	-	3.26	1.41	4.61
3	Mixed	0.831	19.6	0.149	492	4.52	-	3.11	1.51	4.69
10	Mixed	0.838	20.4	0.158	355	3.27	-	2.66	1.69	4.49

30	Excimer	0.837	20.7	0.157	395	3.58	0.27	2.70	1.62	4.37
80	Excimer	0.794	18.6	0.131	1770	15.7	0.27	3.76	1.20	4.51

”

4. Authors should explain why the PLQY of TpAT-tFFO decreased a lot when the excimer formed, but TpATtBu-tFFO excimer showed an insignificant drop in PLQY.

We do not have a clear explanation of the different change in PLQY from monomer to excimer emission, yet. We speculate carefully that, in the intramolecular CT state responsible for the monomer emission, the bulky diphenyltriazine group in TpAT-tFFO is relatively rigid because the donor and acceptor groups are covalently linked via triptycene. However, in the intermolecular CT state responsible for the excimer emission, the void space between molecules allows multiple vibrational and rotational channels for the diphenyltriazine group, thus significantly reducing the PLQY of the CT excimer emission.

In contrast, the smaller di-tert-butyl triazine group in TpATtBu-tFFO makes this effect less significant in the intermolecular CT state because this structure does not substantially increase the vibration and rotation channels even with the void space. This may result in a more stable PLQY for the TpATtBu-tFFO excimer.

However, in our view an understanding of the different PLQY trends for TpAT-tFFO and TpATtBu-tFFO is not central to our current work. In addition, at the present stage it is impossible for us to experimentally confirm our hypothesis on the causes underlying this difference. Therefore, we decided not to include this argument in the revised manuscript.

5. The temperature-dependent PL study should be performed on monomer and excimer state to confirm the TADF properties.

We have now conducted an analysis of the temperature dependence of k_{ISC} and k_{RISC} rates for both TpAT-tFFO and TpATtBu-tFFO by performing variable-temperature transient PL measurements. The measurements were performed across a temperature range between 240K and 300K in order to ensure that the sample remained above the freezing points of acetonitrile (227.1 K) and toluene (178.2 K). Within this temperature range, both materials demonstrated TADF properties in both their monomer (0.1 mM solution) and excimer states (10 mM solution) as expected. We have included these new results in the Supplementary Information and have revised the manuscript as follows:

Line 135 on page 7,

“Transient PL measurements indicate TADF is present across all concentrations tested here (Fig. 1f). **Furthermore, variable-temperature PL shows the increased ISC and RISC rates at higher temperature for both monomer and excimer emissions (see Supplementary Fig. 2).**”

Line 248 on page 12,

“Like TpAT-tFFO, TpATtBu-tFFO exhibited a concentration-dependent transition from monomer to excimer emission, shifting from deep-blue monomer emission at 442 nm to greenish-blue excimer emission at 484 nm (Fig. 4a). **The presence of TADF was confirmed for**

both monomer and excimer emission using variable-temperature time-resolved PL measurements (see Supplementary Fig. 9).”

Supplementary Figs. 2 and 9,

“

“

Supplementary Fig. 2. TADF analysis of TpAT-tFFO solution. Variable-temperature photoluminescence for **a** monomer emission from a solution at 0.1 mM and **b** excimer emission from a solution at 10 mM. **c,d** Analysis of k_{ISC} , k_{RISC} , and activation energy (E_a) for monomer and excimer emission.

Supplementary Fig. 9. TADF analysis of TpATtBu-tFFO solution. Variable-temperature photoluminescence for **a** monomer emission from a solution at 0.1 mM and **b** excimer emission from a solution at 10 mM. **c,d** Analysis of k_{ISC} , k_{RISC} , and E_a for monomer and excimer emission.

”

We have also supplemented the Methods section as follows:

“

PL and absorption measurement

...

Variable temperature photoluminescence measurements were performed on the FluoTime 250 spectrometer coupled with a cryostat (either OptistatDN, Oxford Instruments or CoolSpek UV USP-203, Unisoku). Solutions were placed in a quartz cuvette with 3 mL volume. The sample chamber was filled with helium gas.

”

6. In ECiHF device, the ratio of TBRb to sensitizer is 1:8, which is ~ 10 mol%. How about the other ratio? In HF-OLEDs, a slightly change in the doping concentration of terminal emitter always results in different device performance.

In general, we expect the following correlation between the (relative) concentration of terminal emitter and the device performance: (1) At low concentrations, the FRET rate is reduced, leading to incomplete energy transfer. (2) At high concentration, more terminal emitter molecules participate in the redox reaction, which is undesirable in ECiHF systems. (3) At very high concentrations, self-absorption also becomes significant. These three effects lead to an optimal intermediate concentration of terminal emitter.

However, we again have to request the reviewer's understanding that we were unable to conduct a full optimization of the TBRb concentration for the current material system, which is due to the experimental limitations listed at the start of our response to reviewer #2. Instead, we have included data from experiments using exciplex (rather than excimer) based ECLDs that we reported earlier (Moon, *Advanced Materials*, 2023, 35, 2302544). Both studies used TBRb as emitter, TBAPF₆ as electrolyte, and a toluene-acetonitrile mixture as solvent. However, the exciplex device contains 30 mM of TAPC and 30 mM of TPBi instead of the 80 mM of TpAT-tFFO.

Figure R3a below shows photoluminescence spectra of solutions containing 30 mM TAPC and 30 mM TPBi and varying concentrations of TBRb (reproduced from *Advanced Materials*, 2023, 35, 2302544). This measurement used 373 nm wavelength light to excite TAPC, TPBi and TBRb together. The TAPC:TPBi exciplex emission is at 525 nm, while TBRb emission appears in the 565-585 nm range. As the TBRb concentration increases, the relative emission by the exciplex decreases, and a red shift caused by self-absorption of the TBRb emission is observed. At 10 mM, the exciplex emission is completely suppressed, indicating complete energy transfer to TBRb at this concentration.

In devices, we tested TBRb concentrations of 4 mM, 10 mM, and 20 mM, keeping both TAPC and TPBi at 30 mM. The best performance (highest luminance) was found at 10 mM TBRb, as shown in Figure R3b.

Figure R3. a Photoluminescence of the mixed toluene-acetonitrile solutions containing 30 mM TAPC and TPBi with varying TBRB concentration. **b**, IVL data of ECLDs operating based on exciplex formation with TBRb concentrations of 4 mM, 10 mM, and 20 mM, respectively. The thickness of the liquid layer is 30 μm and the active area is 4 mm^2 .

For both TpAT-tFFO and TpATtBu-tFFO, we also found that efficient energy transfer to TBRb occurs at a TBRb concentration of 10 mM as already shown in Supplementary Figure 3. Although we did not perform an optimization of the TBRb concentration with the excimer system, it is expected to follow a similar trend as the exciplex system. Based on previous device data, even if the optimal concentration is not exactly 10 mM, dramatic changes in ECLD performance with excimer:emitter concentration ratio are unlikely.

7. In Figure 2b, the emission spectrum of glass-mirror device is narrower than that of glass-glass device. Authors should explain this observation.

The glass–glass and glass–mirror devices were identical in terms of electrical and electrochemical conditions; the only difference was that for the latter device, the surface of the bottom glass substrate was coated with a 120 nm silver mirror and a 50 nm passivation layer. The narrower ECL spectrum observed in the glass–mirror device is attributed to self-absorption of TBRb at wavelengths shorter than 570 nm while reflected light passes through a 30 micrometer-thick liquid layer. We have revised Figure 2b to include the absorption spectrum of TBRb, which partially overlaps with the ECL spectra, confirming that the increased self-absorption contributes to the spectral change as follows:

Figure 2b,

“

”

We have also revised the text as follows:

Line 166 on page 8,

“Fig. 2b shows the ECL spectra of these two devices; both spectra peak at 570 nm, clearly indicating that light emission is exclusively from TBRb. By contrast, devices without TBRb showed direct ECL from the electrogenerated CT excimer; see Supplementary Fig. 4. **The glass-mirror device has increased self-absorption by TBRb during the light recycling due to a partial overlap between the ECL spectrum and the absorption spectrum of TBRb. As a result, the emission intensity of the device at wavelengths shorter than the peak wavelength is slightly reduced, leading to a narrower emission spectrum.** ~~The glass-mirror devices showed a slightly reduced emission intensity at wavelengths shorter than the peak wavelength due to self-absorption by TBRb during light recycling.~~”

It is in fact certain that the lower-than-expected brightness in the glass-mirror device is due to self-absorption in this device. Therefore, we have deleted the word "might" from the following sentence:

Line 186 on page 9,

“Self-absorption by TBRb might further contributes to the lower-than-expected brightness in the glass-mirror device.”

8. In HF-OLEDs, beyond FRET from high-energy sensitizer to low-energy terminal emitter, there is a possibility to up-convert/blue-shift the emission (see. ref. *Nat. Photonics* 2024, 18, 554; *Nat. Photonics* 2024, 18, 516 and *Nat. Photonics* 2021, 15, 203). I wonder if it is possible in ECIHF devices.

This is a very interesting and important question. We can share with the reviewer that we have observed signs of such an up-conversion, i.e., FRET from the energy donor to a terminal emitter at a higher energy. This is part of a follow-up study that we are currently conducting, and we kindly request the reviewers understanding that we are currently unable to share experimental data on this

9. For ECL solution, a supporting electrolyte is added to enhance the conductivity of the device. Supporting electrolyte always affects the redox couples in cyclic voltammetry measurement. What is the concentration effect of this electrolyte on the ECL performance? And how about the counter anion effect, e.g. tetrabutylammonium perchlorate instead of tetrabutylammonium hexafluorophosphate? What is the resistance of the solution of the ECL device?

Again, due to experimental limitations, we have included comparative experiments on device performance with varying electrolyte concentrations for ECLDs operating based on the TAPC:TPBi exciplex to address this question.

We have tested devices at electrolyte concentrations of 25, 50, 100, and 200 mM, respectively. Figures R4a and R4b show minimal deviations in device current density and resistance, respectively, across these electrolyte concentrations. Below 2 V, the device resistance ranged from 1.0 kOhm to 2.1 kOhm due to a capacitive current; above 2V, it dropped due to Faradaic processes, reaching about 380 Ohms at 6 V. Similarly, blank devices containing 100 mM and 200 mM of electrolyte, but no organic semiconductor materials, showed minimal variance in current density and resistance between them (magenta and dark yellow curves).

Figure R4. IVL of ECLDs and blank devices with varying electrolyte concentrations. **a**, Current density-voltage curves and **b**, resistance of emissive devices containing active organic materials as well as blank devices containing only the electrolyte. The thickness of the liquid layer was 30 μm and the active area was 16 mm². **c**, Luminance-voltage curves of the emissive devices.

Consequently, we concluded that electrolyte concentrations above 25 mM are generally sufficient to form electric double layers and induce the Faradaic process. However, the concentration of electrolyte still affects the diffusivity of materials in the solution, which can affect ECL intensity and the optimal frequency in an AC-driven device (Lee, *ACS Applied Materials & Interfaces*, 2018, 10, 41562). We have identified 100 mM as the optimal electrolyte concentration for balancing redox rates and diffusion, given that the device shows the highest ECL intensity at this concentration, as shown in Figure R4c. Thus, we applied this electrolyte concentration to the TpAT-tFFO and TpATtBu-tFFO devices investigated in the current manuscript.

Tetrabutylammonium perchlorate has been used about as frequently as tetrabutylammonium hexafluorophosphate as supporting electrolyte in ECL studies. We have not tested tetrabutylammonium perchlorate, but, we agree that the smaller size of the perchlorate ion compared to the hexafluorophosphate ion might enhance ion pairing and rates of the redox process. However, its narrower electrochemical stability window is likely to compromise the long-term operational stability of the device. We would like to defer further tests of this to a future study, given it is not essential to the current work.

Reviewer #3 (Remarks to the Author):

This paper reports the fundamental device properties of solution-processed electroluminescent devices (ECLDs) based on a newly proposed architecture. The authors achieved notable improvements in device performance, including luminance, efficiency, and operational stability. They also investigated the excimer characteristics of TADF molecules through optical spectroscopy. The use of a hyperfluorescence mechanism in this device architecture is particularly interesting. Although the devices are not yet ready for practical applications, the improvements reported are significant and relevant to the development of other types of electroluminescent devices. From this perspective, I judge that this paper merits publication in Nature Communications. However, the following points should be addressed in the revised version.

We sincerely appreciate the reviewer's valuable comments and the very positive overall assessment of our work. We have written a comprehensive answer to the reviewer's three questions about spectroelectrochemistry below.

- The spectroscopic results shown in Figure 5 are intriguing. However, the interpretation of the data in Figures 5e and 5f is somewhat confusing. The authors attribute the observed signals to ions of TBRb or of the TADF molecule. This assignment is questionable, as the expected absorption features from TADF molecule ions would likely include a strong positive signal in the 460–520 nm region, which appears to be absent.

- In Figure 5d, the ionized TBRb shows both a positive absorption peak at 434 nm and negative bleaching signals at 490 nm and 525 nm. In contrast, Figure 5e shows that the positive signal near 430 nm is significantly weaker than in the TBRb ion spectrum, while the bleaching signals at 490 nm and 525 nm remain similar in intensity. This suggests that the bleaching cannot be attributed solely to the ionized species. Instead, the bleaching might partly originate from luminescent excitons of TBRb, which is plausible given TBRb's long-lived excited state, as shown in Figure 1h.

- Notably, the intensity difference in the positive signal between the 190Hz and 1kHz conditions may reflect a difference in emission intensity. Indeed, Figure 5e suggests stronger emission at 190 Hz than at 1 kHz, consistent with the left panel of Figure 4b (despite the difference in applied voltage). However, the data in Figure 5f does not align with this interpretation. In particular, the pronounced signal around 580 nm suggests the involvement of an additional mechanism. While this is just one possible interpretation, the current explanation remains ambiguous. A clearer and more consistent explanation should be included in the revised manuscript.

We acknowledge that the spectroelectrochemical experiments were not explained/analyzed as well as we would have liked. After a thorough review of the data and performing additional measurements, we have now revised Fig. 5 with a new panel and have completely updated the discussion. Importantly, our new measurements can now differentiate the signal from the respective cationic and anionic absorption bands of each molecule. With this new data, we can now explain our observations more convincingly. While the reviewer's interpretation is certainly interesting, the new data shows a slightly different picture. We would like to ask the reviewer to find the new Figure 5 and updated discussion below

New panels 5a, 5b, and 5c show the ΔOD results of TpAT-tFFO, TpATtBu-tFFO, and TBRb under DC operation, which was plotted together in old panel 4d. The graphs now include the

signals of their respective cationic and anionic absorption bands that we have newly measured. The schematic for the spectroelectrochemistry setup, originally shown in panel 5a, has been relocated to the Supplementary Fig. 12a, where we have also included the setup for resolving ion absorption bands using the devices in a coplanar electrode array.

The figures and discussion have been updated as follows:

Figures 5a-c,

“

Fig. 5. Absorption spectroelectrochemistry. Measured absorption spectra of **a** TpAT-tFFO, **b** TpATtBu-tFFO, and **c** TBRb ions under DC operation of devices in parallel-electrode configuration at 3.5 V, 3.5 V, and 3.0 V, respectively. The absorption bands are further resolved in coplanar-electrode configuration.

”

Line 278 on page 15,

"To gain deeper insights into the differences between TpAT-tFFO and TpATtBu-tFFO devices, we performed spectroelectrochemistry analysis³⁶ (see the setup in the Methods section and Supplementary Fig. 12). This technique tracks changes in optical density (ΔOD) as light passes through the active area of the ECLD with and without voltage application. When applying DC voltage to the TpAT-tFFO device without the terminal emitter TBRb (Fig. 5a), an absorption peak for TpAT-tFFO cations was observed at 471 nm, along with a broad absorption band for anions ranging from 420 to 620 nm. The TpATtBu-tFFO device exhibited similar absorption bands, as shown in Fig. 5b, attributed to its structural similarity. For a device using solely TBRb (Fig. 5c), absorption peaks were detected at 424 nm for cations and 438 nm for anions, respectively. The bleaching of TBRb absorption at 490 nm and 525 nm ($\Delta OD < 0$) is attributed to the reduced number of neutral TBRb molecules due to ionization."

For the AC response of TpAT-tFFO devices (originally shown in panel 5e), we rearranged the data in panel 5d to include a simulation of ΔOD using a superposition of the ion absorption bands shown in panels 5a and 5c, along with the negative TBRb absorption band to simulate bleaching. This simulation helps to identify the types of accumulated ions and the contribution of TBRb excitons to bleaching. Unlike in the original version of our manuscript, we did not normalize the ΔOD values to facilitate a direct comparison of ion and exciton generation at different operating frequencies. Importantly, while signatures of TBRb(+) and TBRb(-) ions as well as bleaching of the neutral TBRb absorption band are all seen in the spectrum, there is no indication that TpAT-tFFO accumulate.

To illustrate this more clearly, the schematic of the operational mechanism (originally Fig. 5b) has been updated (now Fig. 5e) to emphasize ions that accumulate during operation versus those that do not.

The revised figures and discussion are as follows:

Figures 5d-e,

“

d Spectroscopic analysis of optical density changes in TpAT-tFFO devices under AC operation at frequencies of 190 Hz and 1 kHz. The simulations use the absorption spectra of TBRb(+) and TBRb(-) ions, along with bleaching of the neutral TBRb absorption band. **e** Schematic of the ECL process based on excimer-formation on TpAT-tFFO and subsequent energy-transfer to TBRb. Due to the rapid nature of this process, TpAT-tFFO ions do not accumulate in the device. However, any TBRb ions formed by a direct redox process at the electrode will accumulate.

”

Line 306 on page 16,

“Fig. 5d shows the ΔOD curves of the TpAT-tFFO device under AC operation at $V_{rms} = 3.5$ V and frequencies of 190 Hz and 1 kHz (green and yellow lines), respectively. We simulated the ΔOD curves (blue and brown broken lines) using the cationic and anionic absorption spectra of TpAT-tFFO and TBRb, as well as the bleaching spectrum of neutral TBRb. At both frequencies, the ΔOD simulations include only the combined absorption from TBRb(+) and TBRb(-) along with TBRb bleaching, while no absorption from TpAT-tFFO(+) or TpAT-tFFO(-) was observed. This indicates that, as shown schematically in Fig. 5e, TpAT-tFFO ions swiftly return to their neutral state via rapid excimer formation followed by a swift FRET process, thus bypassing unnecessary electron transfer processes. This operation produces TBRb(+) and TBRb(-) by redox processes at the electrode surfaces. These ions tend to accumulate during device operation as the annihilation process between them is slower than the excimer formation. The formation of these long-lived TBRb excitons leads to bleaching of the TBRb absorption bands. The larger degree of bleaching observed at 190 Hz, compared to 1 kHz, is attributed to the increased formation of TBRb excitons, consistent with the frequency-luminance curve in Fig. 3b.”

A similar analysis was performed for the TpATtBu-tFFO device, and the results are plotted in new Fig. 5f. Here, the absorption band observed at wavelengths around 580 nm is clearly attributed to TpATtBu-tFFO(-), while the peak at 424 nm corresponds to TBRb(+). Interestingly, TpATtBu-tFFO(+) and TBRb(-) were not detected. In our understanding, this is likely due to rapid electron exchange between the TBRb molecule and TpATtBu-tFFO(+) ion, is schematically illustrated in new panel 5g.

The revised figures and discussion are as follows:

Figures 5f-g

“

f Spectroscopic analysis of optical density changes in TpATtBu-tFFO devices. **g** Schematic of the electron transfer processes, resulting in accumulation of TBRb(+) and TpATtBu-tFFO(-). The significant gap of 0.55 eV between the HOMO levels of TBRb and TpATtBu-tFFO facilitates the HOMO-to-HOMO electron transfer process.

”

Line 321 on page 16,

“The simulation results for ΔOD of the TpATtBu-tFFO device (Fig. 5f), operating at $V_{rms} = 3.5$ V, show an accumulation of TBRb(+) and TpATtBu-tFFO(-) ions along with TBRb bleaching at both tested frequencies. This indicates TBRb(-) and TpATtBu-tFFO(+) do not accumulate during device operation. As illustrated in the schematic in Fig. 5g, the different shape of the ΔOD curve is likely because of a HOMO-to-HOMO electron transfer from neutral TBRb to TpATtBu-tFFO(+) that proceeds at a rate comparable to or faster than the excimer formation. This leads to an overproduction of TBRb(+) and insufficient generation of TpATtBu-tFFO(+), which in turn results in an imbalance in ion concentration; TBRb(-) and TpATtBu-tFFO(+) are effectively depleted through annihilation and excimer formation, while their corresponding counter ions accumulate during device operation. Our spectroelectrochemistry data therefore confirms the mixed operation mechanism of the TpATtBu-tFFO device. It can further be seen that the enhanced excimer formation process at higher frequencies, shown in the current-voltage-luminance curves in Fig. 4c, correlates with a decreased ΔOD from TpATtBu-tFFO(-) at a higher frequency.

The electron transfer process is facilitated by a significant energy level difference ΔE_{HH} between the HOMO levels of TpATtBu-tFFO and TBRb, namely $\Delta E_{HH} = 0.55$ eV. By contrast, electron transfer between TpATtBu-tFFO and TBRb was not significant, likely due to the smaller ΔE_{HH} of 0.39 eV. Similarly, the LUMO-to-LUMO electron transfer is not likely to be significant between TpATtBu-tFFO and TBRb as the energy level difference ΔE_{LL} is only 0.29 eV.

Although ΔE_{LL} between TpATtBu-tFFO and TBRb is also large at 0.52 eV, we did not observe any evidence of LUMO-to-LUMO electron transfer in our experiments.”

The conclusion has been updated as follows to include our new findings:

Line 363 on page 18,

“In contrast, the device using TpATtBu-tFFO, where ΔE_{HH} and ΔE_{LL} exceed exceeds 0.5 eV, showed S-shaped and frequency dependent JVL curves and an upward shift in the optimal frequency with increasing voltage, both likely arising Both likely arise from a mixed operating mechanism involving excimer formation and ionic annihilation, whose ratio varies with applied voltage and frequency. Spectroelectrochemical analysis further revealed that at higher frequencies the predominant operating mechanism in the TpATtBu-tFFO device is excimer formation and subsequent energy transfer, whereas electron transfers and subsequent ionic annihilation appear to dominate at lower frequencies. Spectroelectrochemical analysis further attributed the mixed operating mechanism to an imbalance in ion concentrations facilitated by HOMO-to-HOMO electron transfer from neutral TBRb to TpATtBu-tFFO(+) ions.”

The Method section has been updated as follows:

“Absorption spectroelectrochemistry

Spectroelectrochemical analysis was performed using a custom-built inverted microscope setup (Nikon Eclipse Ti2) equipped with 20x extra-long working distance air objective. A schematic of the measurement setup is given in Supplementary Fig. 12a. The sample was illuminated by a white LED source (pE-4000, CoolLED). A beam splitter and a digital camera (C13440 Orca-Flash 4.0, Hamamatsu) ensured precise alignment of the observation area within the active region of the device. The transmitted light spectrum was recorded using a spectrometer (Andor Shamrock 500i) with a 1-second acquisition time. First, the transmission of the white LED was measured with the device turned off (I_{off} in Fig. 5a), followed by a measurement with the device turned on (I_{on}). Lastly, the ECL intensity was measured with the white LED switched off (I_{ECL}). ΔOD is calculated as $\log_{10}\{I_{off}/(I_{on}-I_{ECL})\}$. The absorption of TpAT-tFFO and TpATtBu-tFFO ions was measured by applying a 3.5 V DC voltage to each TpAT-tFFO and TpATtBu-tFFO device used in Fig. 5e-f. The absorption of TBRb ions was measured using a device with a solution containing 10 mM TBRb with a supporting electrolyte, with application of 3.0 V DC voltage. Ion absorption bands for TpAT-tFFO, TpATtBu-tFFO, and TBRb were measured using toluene:acetonitrile solutions containing 80 mM of TpAT-tFFO, 80 mM of TpATtBu-tFFO, and 10 mM of TBRb, respectively. Each solution included 100 mM of supporting electrolyte. The solution in each device had a thickness of 30 μm thickness and devices were operated under DC voltage. Devices in parallel and coplanar electrode configuration, as shown in Supplementary Fig. 12b, were used to analyze the overall ion absorption profile and the shapes of the cation and anion absorption bands, respectively. for the simulations of the ΔOD spectra under AC driving, the ratio of each absorption band measured in DC mode was adjusted to match the overall absorption profile.”

Supplementary Fig. 12, including a schematic for the spectroelectrochemistry setup, has newly been added to Supplementary Information as follows:

“

Supplementary Fig. 12. a Schematic of the absorption spectroelectrochemistry setup. **b** Device configurations, operating methods, and fields of view (FOVs) used to analyze the ion absorption bands. Under AC operation significant electroluminescence (ECL) is generated, whereas DC operation involves little ECL. For the parallel electrode configuration, cations and anions coexist within the FOV, while the coplanar electrode configuration allows to measure the absorption bands of each cation and anion separately.

”